# A global surface $CO_2$ flux dataset (2015–2022) inferred from OCO-2 retrievals using the GONGGA inversion system

Zhe Jin[1,2], Xiangjun Tian[1,3], Yilong Wang[1], Hongqin Zhang[4], Min Zhao[1], Tao Wang[1], Jinzhi Ding[1], Shilong Piao[1,2,1]

[1]State Key Laboratory of Tibetan Plateau Earth System, Resources and Environment (TPESRE), Institute of Tibetan Plateau Research, Chinese Academy of Sciences, Beijing, 100101, China
[2]Institute of Carbon Neutrality, College of Urban and Environmental Sciences, Peking University, Beijing, 100871, China
[3]University of Chinese Academy of Sciences, Beijing, 101408, China
[4]Institute of Atmospheric Physics, Chinese Academy of Sciences, Beijing, 100029, China

*Correspondence to:* Xiangjun Tian (tianxj@itpcas.ac.cn) and Yilong Wang (wangyilong@itpcas.ac.cn)

**Abstract.** Accurate assessment of the size and distribution of carbon dioxide ($CO_2$) sources and sinks is important for efforts to understand the carbon cycle and support policy decisions regarding climate mitigation actions. Satellite retrievals of the column-averaged dry-air mole fractions of $CO_2$ ($XCO_2$) have been widely used to infer spatial and temporal variations of carbon fluxes through atmospheric inversion techniques. In this study, we present a global spatially resolved terrestrial and ocean carbon flux dataset for 2015–2022. The dataset was generated by the Global ObservatioN-based system for monitoring Greenhouse GAses (GONGGA) atmospheric inversion system through the assimilation of Orbiting Carbon Observatory 2 (OCO-2) $XCO_2$ retrievals. We describe the carbon budget, interannual variability, and seasonal cycle for the global scale and a set of TransCom regions. The 8-year mean net biosphere exchange and ocean carbon fluxes were $-2.22 \pm 0.75$ Pg C $yr^{-1}$ and $-2.32 \pm 0.18$ Pg C $yr^{-1}$, absorbing approximately 23% and 24% of contemporary fossil fuel $CO_2$ emissions, respectively. The annual mean global atmospheric $CO_2$ growth rate was $5.17 \pm 0.68$ Pg C $yr^{-1}$, which is consistent with the National Oceanic and Atmospheric Administration (NOAA) measurement ($5.24 \pm 0.59$ Pg C $yr^{-1}$). Europe has the largest terrestrial sink among the 11 TransCom land regions, followed by Boreal Asia and Temperate Asia. The dataset was evaluated by comparing posterior $CO_2$ simulations with Total Carbon Column Observing Network (TCCON) retrievals as well as Observation Package (ObsPack) surface flask observations and aircraft observations. Compared with $CO_2$ simulations using the unoptimized fluxes, the bias and root mean square error of posterior $CO_2$ simulations were largely reduced across the full range of locations, confirming that the GONGGA system improves the estimates of spatial and temporal variations in carbon fluxes by assimilating OCO-2 $XCO_2$ data. This dataset will improve the broader understanding of global carbon cycle dynamics and their response to climate change. The dataset can be accessed at https://doi.org/10.5281/zenodo.8368846 (Jin et al., 2023a).

**Keywords: global carbon cycle, atmospheric $CO_2$, atmospheric inversion, $CO_2$ fluxes, Observing Carbon Observatory 2, interannual variability, seasonal cycle**

# 1    Introduction

Atmospheric carbon dioxide ($CO_2$) concentrations are rapidly rising, mainly because of increases in anthropogenic emissions. Land and oceans can absorb substantial amounts of $CO_2$ and thus mitigate global warming. During the past decade (2012–2021), approximately one fourth of total $CO_2$ emissions were absorbed by the land and oceans, respectively (Friedlingstein et al., 2022). However, there are large uncertainties in estimates of the size, spatial distribution, and interannual variability of land and ocean fluxes (Piao et al., 2009b; Eldering et al., 2017; Hauck et al., 2020; Piao et al., 2020). Accurate estimates of these fluxes at global and regional scales are essential for improving overall knowledge regarding the current status of the carbon cycle and projecting long-term changes (Zscheischler et al., 2017).

There are many methods for the estimation of global and regional carbon budgets, including the inventory method, the eddy covariance method, the ecosystem process modelling method, and the atmospheric inversion method (Piao et al., 2022). The first three methods upscale the site-level ground observations using statistical or process-based models; they are usually regarded as bottom-up approaches. In contrast, atmospheric inversion infers carbon fluxes by combining information from atmospheric $CO_2$ concentrations, prior flux estimates, and atmospheric transport (Bousquet et al., 2000; Gurney et al., 2002), which is regarded as a top-down approach. Atmospheric inversion is appropriate for assessments of global and regional carbon fluxes because spatiotemporal variations in atmospheric $CO_2$ concentrations contain the signatures of sources and sinks at large spatial scales. However, inversion accuracy is limited by the numbers and distributions of atmospheric $CO_2$ observations, uncertainties regarding the atmospheric transport model and the $CO_2$ emission inventories (such as fossil fuel combustion emissions), and insufficient knowledge of prior flux uncertainties (Liu et al., 2021; Piao et al., 2022).

Currently, atmospheric inversions use either ground-based or space-based observations. Ground-based in situ and flask observations have higher precision, but they are unevenly distributed. Most ground-based observations are mainly concentrated in North America and Europe (Peters et al., 2007; Chevallier et al., 2010; Lauvaux et al., 2016). Inversions using in situ and flask observations can consistently constrain surface $CO_2$ fluxes at the global scale and at some regional scales (for well-sampled continents), but their uncertainty rapidly increases at the sub-continental scale or when considering continents with sparse observations (Peylin et al., 2013; Byrne et al., 2017; Crowell et al., 2019). For example, there are only eight sites in the Chinese mainland under the World Meteorological Organization/Global Atmosphere Watch program (Wang et al., 2020b), and Chinese land sinks constrained by in situ $CO_2$ observations can differ by up to an order of magnitude (Chen, 2021; Wang et al., 2022a; Wang et al., 2022b). The space-based column-averaged $CO_2$ dry-air mole fraction ($XCO_2$) retrievals serve as an emerging data stream for atmospheric inversions. Satellite $XCO_2$ retrievals have broader spatial coverage than in situ and flask observations; accordingly, they fill observational gaps over areas with few stations. The two most widely used satellites dedicated to measure $CO_2$ are Greenhouse gases Observing SATellite (GOSAT) (Yokota et al., 2009) and Orbiting Carbon Observatory 2 (OCO-2) (Crisp et al., 2004). GOSAT retrievals have been used in multiple inversions and were shown to be able to reduce the uncertainty of flux estimates in regions where surface $CO_2$ observations are sparse (Takagi et al., 2011; Basu et al., 2013; Chevallier et al., 2014). The OCO-2 team updates satellite

retrievals roughly once per year. Refinements of instrument error characterization, retrieval algorithms, and bias correction procedures have led to substantial improvements in the accuracy and precision of satellite-retrieved $XCO_2$ data through these updates (O'dell et al., 2018; Kiel et al., 2019); the single sounding random error of official OCO-2 retrievals is now better than 1 ppm (Eldering et al., 2017; Wunch et al., 2017). These improvements in $XCO_2$ retrievals have a transformative effect on satellite-based estimates of global carbon fluxes (O'dell et al., 2018; Miller and Michalak, 2020). For example, the OCO-2 version 7 retrievals—the basis of early inversion studies using OCO-2 data—are fit to constrain land carbon fluxes at continental and hemispheric scales (Miller et al., 2018; Crowell et al., 2019). Chevallier et al. (2019) showed that the OCO-2 version 9 retrievals have similar performance in terms of constraining carbon fluxes to the inversions that use observations from surface stations when the inversed fluxes and $CO_2$ concentrations are compared with independent aircraft data. More recently, the OCO-2 team has released the retrieval product for version 11r. The effectiveness and potential applications of these updated satellite retrievals in efforts to infer surface $CO_2$ fluxes require continuous and persistent investigation.

In this study, we used the GONGGA (Global ObservatioN-based system for monitoring Greenhouse GAses) inversion system (Jin et al., 2023b) to generate a global dataset of terrestrial ecosystem and ocean carbon fluxes from 2015 to 2022 by assimilating OCO-2 $XCO_2$ retrievals (v11r). Here, we present the prior and posterior global 3-hourly gridded terrestrial ecosystem and ocean carbon fluxes at a spatial resolution of 2° latitude × 2.5° longitude. Gridded fluxes from fossil fuel emissions and biomass burning emissions are also available for inferring the total fluxes.

This paper is organized as follows: section 2 describes the methods and data used; section 3 describes the format and content of the dataset; section 4 analyzes the key characteristics of global and regional carbon cycles; section 5 evaluates posterior fluxes using TCCON and ObsPack observations; section 6 introduces data availability; and section 7 summarizes the paper.

## 2 Methods and Data

### 2.1 The GONGGA inversion system

GONGGA is an atmospheric inversion system that constrains gridded carbon fluxes with atmospheric $CO_2$ observations and transport simulations (Jin et al., 2023b). The assimilated observations are OCO-2 v11r $XCO_2$ retrievals (OCO-2/OCO-3 Science Team et al., 2022), and the transport model is GEOS-Chem v12.9.3 (Suntharalingam et al., 2004; Nassar et al., 2010; Nassar et al., 2013). The spatial resolution of GEOS-Chem is 2° latitude × 2.5° longitude, with 47 layers in the vertical direction from the surface to the top of the atmosphere. The model is driven by Modern-Era Retrospective analysis for Research and Applications 2 (MERRA-2) meteorological data provided by the Goddard Earth Observing System (GEOS) of the National Aeronautics and Space Administration (NASA) Global Modeling and Assimilation Office (Gelaro et al., 2017). Four types of carbon fluxes are used to drive the atmospheric $CO_2$ simulations, including NEE (net ecosystem exchange, i.e., the balance of photosynthesis and respiration) from terrestrial ecosystems, atmosphere-ocean carbon exchange, fossil fuel carbon emissions, and biomass burning carbon emissions. NEE and ocean carbon fluxes are optimized by GONGGA,

whereas fossil fuel emissions and biomass burning emissions are assumed to be well-known and not optimized, which is a usual convention in global atmospheric inversions (Peters et al., 2007; Jiang et al., 2022).

GONGGA uses the nonlinear least squares four-dimensional variational data assimilation (NLS-4DVar) method (Tian and Feng, 2015; Tian et al., 2018) to minimize the following cost function:

$$J(\boldsymbol{x}) = \frac{1}{2}(\boldsymbol{x} - \boldsymbol{x}_a)^{\mathrm{T}} \mathbf{B}^{-1}(\boldsymbol{x} - \boldsymbol{x}_a) + \frac{1}{2}\big(\boldsymbol{y} - H(\boldsymbol{x})\big)^{\mathrm{T}} \mathbf{R}^{-1}\big(\boldsymbol{y} - H(\boldsymbol{x})\big). \tag{1}$$

where $\boldsymbol{x}$ is the state vector that contains the variables to be optimized and $\boldsymbol{x}_a$ is its prior estimate; $\boldsymbol{y}$ gathers the $XCO_2$ retrievals; $\mathbf{B}$ is the prior error covariance matrix and $\mathbf{R}$ is the observation error covariance matrix; $H(\cdot)$ is the observation operator, which relies on GEOS-Chem simulations and sampling of modelled atmospheric $CO_2$. Firstly, the atmospheric transport model is used to simulate gridded $CO_2$ concentrations driven by surface fluxes. Then, the simulated gridded $CO_2$ profiles are interpolated horizontally by inverse distance weighting and vertically by linear interpolation on pressure. Thirdly, the interpolated $CO_2$ profiles are used to construct the simulated $XCO_2$ using the equation:

$$XCO_2^m = XCO_2^a + \boldsymbol{h}^{\mathrm{T}} \mathbf{A}\big(\boldsymbol{x}_{\mathrm{CO2}} - \boldsymbol{x}_{\mathrm{CO2},a}\big). \tag{2}$$

where $XCO_2^m$ is the modelled $XCO_2$, $\boldsymbol{x}_{\mathrm{CO2}}$ is the interpolated $CO_2$ profile from the GEOS-Chem simulation. $XCO_2^a$, $\boldsymbol{h}$, $\mathbf{A}$, and $\boldsymbol{x}_{\mathrm{CO2},a}$ are the prior value of $XCO_2$, the pressure weighting function, the averaging kernel matrix, and the prior $CO_2$ vertical profile, respectively, provided by the OCO-2 Lite file.

The optimization algorithm NLS-4DVar is a hybrid method that combines the advantages of the conventional four-dimensional variational (4D-Var) method and ensemble Kalman filter (EnKF), achieving high inversion accuracy with low computational cost and complexity (Tian and Feng, 2015; Tian et al., 2018). GONGGA adopts a novel dual-pass inversion strategy, successively optimizing initial $CO_2$ concentrations and surface carbon fluxes within each inversion window of 14 days, which distinguishes model–data mismatches caused by errors from initial $CO_2$ concentrations and surface fluxes (Jin et al., 2023b). Note that during the flux optimization, the state vector $\boldsymbol{x}$ gathers gridded scaling factors for NEE and ocean carbon fluxes. The spatial resolutions of the optimization for both initial $CO_2$ concentrations and fluxes are 2° latitude × 2.5° longitude, the same as the transport model resolution. The temporal resolution of the optimization is 14 days, indicating that the fluxes within each 14-day window are uniformly adjusted by the same scaling factor. In this study, GONGGA was run from September 6, 2014, to December 31, 2022. The 2014 results were regarded as spin-up, whereas the 8-year results spanning 2015–2022 comprised the dataset.

## 2.2 Prior CO₂ fluxes and uncertainties

The prior $CO_2$ fluxes include NEE, ocean-atmosphere carbon fluxes, fossil fuel emissions, and biomass burning emissions. The prior NEE was simulated by ORCHIDEE-MICT (Guimberteau et al., 2018). The prior ocean carbon fluxes were from the CT2022 $p$CO₂-Clim prior data, which were derived from the Takahashi et al. (2009) climatology of seawater $p$CO₂. Fossil fuel emissions were from the monthly Global Carbon Budget Gridded Fossil Emissions Dataset (GCP-GridFED; version 2023.1) (Jones et al., 2021). Biomass burning emissions were from the Global Fire Emissions Database (GFED,

version 4.1s) 0.25° × 0.25° monthly data scaled with daily factors (Randerson et al., 2017; Van Der Werf et al., 2017). For the estimation of prior flux uncertainties, we first used a prior perturbation ensemble to approximate the prior scaling factor error covariance matrix, then calculated the prior flux uncertainties through the matrix multiplication between the scaling factor error covariance matrix and prior fluxes. The posterior flux uncertainties were calculated in the same manner, using the ensemble of posterior scaling factors and prior fluxes. The difference between the prior and posterior flux uncertainties was regarded as the difference in the perturbation ensemble. For detailed steps, see Text S1 in the Supplement.

## 2.3 OCO-2 column $CO_2$ observations

We used OCO-2 Level 2 Lite v11r $XCO_2$ products (O'dell et al., 2012; O'dell et al., 2018; Gunson and Eldering, 2020) retrieved by the Atmospheric Carbon Observations from Space (ACOS) algorithm (Connor et al., 2008) to constrain the surface carbon fluxes. The OCO-2 satellite carries high-resolution spectrometers that return high-precision measurements of reflected sunlight received within the $CO_2$ and $O_2$ bands in the short-wave infrared spectrum (Crisp et al., 2012). The OCO-2 spacecraft flies in a 705-km-altitude sun-synchronous orbit with a 16-day (233 orbits) ground track repeat cycle. OCO-2 has a footprint of $1.29 \times 2.25$ km$^2$ at nadir mode and acquires eight cross-track footprints, creating a swath width of 10.3 km.

Before assimilation, the $XCO_2$ retrievals were filtered with the xco2_quality_flag parameter provided by the OCO-2 Lite products; xco2_quality_flag = 0 (1) denotes good (bad) retrieval quality. Only retrievals with good quality were selected. We applied a data thinning algorithm (Liu and Rabier, 2002; Campbell et al., 2017; Reale et al., 2018) to reduce the potential impacts of correlated errors in adjacent soundings. We set the threshold of the number of daily observations to 20,000. If the number of good retrievals exceeded the threshold within a single day, excess data were removed. For example, if there were 60,000 good retrievals in one day, one of every three sequential retrievals was selected according to sounding ID. Before data thinning, there were 203,368,424 $XCO_2$ retrievals with good quality from September 6, 2014, to December 31, 2022. After data thinning, 40,337,763 $XCO_2$ retrievals were actually assimilated in the inversion, about a fifth of total good retrievals. Furthermore, OCO-2 retrievals were scaled to the official World Meteorological Organization (WMO) X2019 standards following instructions provided by the National Oceanic and Atmospheric Administration (NOAA, https://docs.google.com/document/d/e/2PACX-1vQ0JqK72fAOThaJwJyILLgfOE2qpHYdgNsIYAs6T2cMGumwVliSK7lurIYKCMOFgz1fyxuKYwlm5FEx/pub, last access: September 12, 2023).

## 2.4 Evaluation of posterior fluxes

Generally, it is difficult to directly verify the posterior fluxes because of the lack of direct flux observations that exhibit a footprint size comparable with the spatial resolution of global inversion models (typically several hundred kilometers). Instead, we compared the simulated $CO_2$ concentrations driven by posterior fluxes with atmospheric $CO_2$ observations to achieve indirect verification (e.g., Wang et al. (2019); Wu et al. (2020); Liu et al. (2021)). In this study, we performed these

comparisons using observations from TCCON version GGG2020 (Laughner et al., 2023) and Obspack ($CO_2$ GLOBALVIEWplus v8.0 and NRT v8.1) datasets (Cox et al., 2022; Di Sarra et al., 2023).

### 2.4.1 TCCON XCO₂ retrievals

TCCON is a network of ground-based Fourier transform spectrometers that record direct solar spectra in the near-infrared spectral region. From these spectra, accurate and precise column-averaged $CO_2$ abundances are retrieved and reported (Wunch et al., 2011). TCCON XCO₂ retrievals are estimated to have precisions better than 0.25% (i.e., ~1 ppm) (Wunch et al., 2011). These retrievals have been used as primary validation data for several satellite missions, including GOSAT and OCO-2 (Wunch et al., 2011; Wunch et al., 2017). Here, we used GGG2020 version data (Wunch et al., 2015). There are 27 TCCON sites with observations covering the inversion period (Table 1); the site locations are shown in Figure 1a.

**Table 1. Geographic locations and references of TCCON sites used for validation. Sites are listed according to latitude from north to south.**

| Station | Latitude | Longitude | Country | Data Reference |
|---|---|---|---|---|
| Eureka | 80.0 | −86.4 | Canada | Strong et al. (2022) |
| Ny Ålesund | 78.9 | 11.9 | Norway | Buschmann et al. (2022) |
| Sodankylä | 67.4 | 26.6 | Finland | Kivi et al. (2022) |
| East Trout Lake | 54.4 | −105.0 | Canada | Wunch et al. (2022) |
| Bremen | 53.1 | 8.9 | Germany | Notholt et al. (2022) |
| Harwell | 51.6 | −1.3 | United Kindom | Weidmann et al. (2023) |
| Karlsruhe | 49.1 | 8.4 | Germany | Hase et al. (2022) |
| Paris | 49.0 | 2.4 | France | Té et al. (2014) |
| Orléans | 48.0 | 2.1 | France | Warneke et al. (2022) |
| Garmisch | 47.5 | 11.1 | Germany | Sussmann and Rettinger (2022) |
| Park Falls | 46.0 | −90.3 | United States | Wennberg et al. (2022d) |
| Rikubetsu | 43.5 | 143.8 | Japan | Morino et al. (2022b) |
| Xianghe | 39.8 | 117.0 | China | Zhou et al. (2022) |
| Lamont | 36.6 | −97.5 | United States | Wennberg et al. (2022b) |
| Tsukuba | 36.1 | 140.1 | Japan | Morino et al. (2022a) |
| Nicosia | 35.1 | 33.4 | Cyprus | Petri et al. (2022) |
| Edwards | 35.0 | −117.9 | United States | Iraci et al. (2022) |
| Jet Propulsion Laboratory | 34.2 | −118.2 | United States | Wennberg et al. (2022a) |

| Pasadena | 34.1 | –118.1 | United States | Wennberg et al. (2022c) |
|---|---|---|---|---|
| Saga | 33.2 | 130.3 | Japan | Shiomi et al. (2022) |
| Hefei | 31.9 | 117.2 | China | Liu et al. (2022) |
| Izana | 28.3 | –16.5 | Spain | García et al. (2022) |
| Burgos | 18.5 | 120.7 | Philippines | Morino et al. (2022c) |
| Manaus | –3.2 | –60.6 | Brazil | Dubey et al. (2022) |
| Réunion Island | –20.9 | 55.5 | France | De Mazière et al. (2022) |
| Wollongong | –34.4 | 150.9 | Australia | Deutscher et al. (2023) |
| Lauder | –45.0 | 169.7 | New Zealand | Pollard et al. (2022); Sherlock et al. (2022) |

### 2.4.2 ObsPack $CO_2$ observations

ObsPack is a framework that combines atmospheric greenhouse gas observations from various sampling platforms (e.g., surface, aircrafts, towers, or ships) and strategies (e.g., flask or in situ), ensuring consistent data quality (Masarie et al., 2014). In this study, we used the surface flask and aircraft observations from obspack_co2_1_GLOBALVIEWplus_v8.0_2022-08-27 in 2015–2021 and obspack_co2_1_NRT_v8.1_2023-02-08 in 2022, both of which are established according to the WMOX2019 scale (Cox et al., 2022). Surface flask observations are usually made on a weekly basis. During the 2015–2022 period, surface flask observations from 57 sites with parameter CT_assim = 1 or 2 were used for evaluation (Fig. 1b). Observations may be reported by multiple institutes at a single site. Here, we only used data from the NOAA laboratory and ignored duplicate records from other sites. Based on the spatial distribution of surface flask sites, we evaluated terrestrial carbon fluxes in six regions: North America, South America, Europe, Africa, East Asia, and Australia (Fig. 1b).

Aircraft observations contain data from the Comprehensive Observation Network for TRace gases by AIrLiner (CONTRAIL) program, the Atmospheric Tomography Mission (ATom), and several localized measurements concentrated in North America (Fig. 1b). The CONTRAIL program is Japan's unique aircraft observation project that measures atmospheric $CO_2$ concentrations using instruments onboard Japan Airlines (JAL) commercial airliners. In the $CO_2$ GLOBALVIEWplus v8.0 ObsPack dataset, the CONTRAIL program contains aircraft measurements between Japan and Australia from 2015 to 2021. The ATom was a NASA Earth Venture Suborbital-2 mission that investigated the greenhouse gases in the atmosphere over the Pacific and Atlantic Ocean from August 2016 to May 2018. We used $CO_2$ observations above the planetary boundary layer (altitude > 1 km) for evaluations to avoid effects on local emissions related to aircraft ascent and descent.

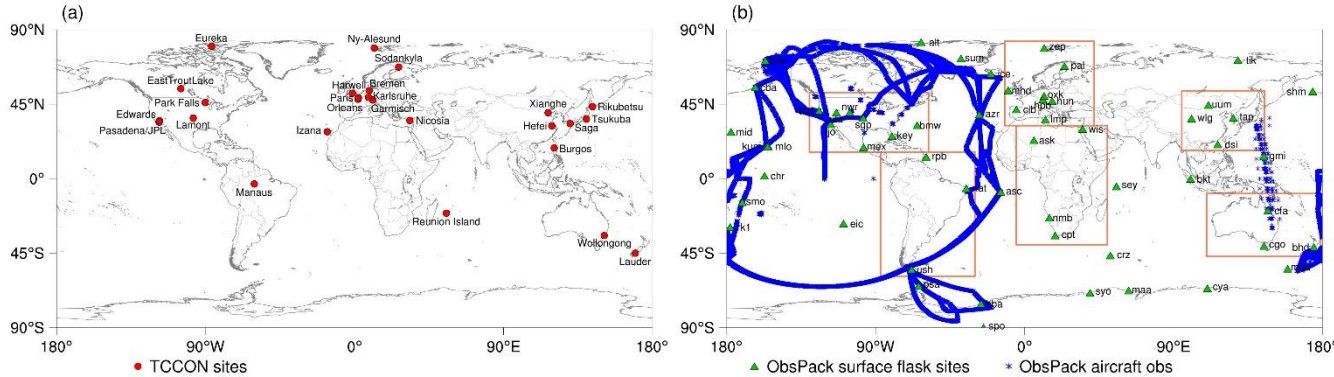

**Figure 1. Spatial distributions of (a) TCCON sites and (b) ObsPack sites used for flux evaluations. Rectangles show the ranges of the six regions used for comparisons with surface flask observations.**

## 3 Dataset description

Here, we present a global dataset that contains surface carbon fluxes from 2015 to 2022. The flux files contain NEE, ocean-atmosphere carbon fluxes, fossil fuel emissions, and biomass burning emissions. The NEE and ocean-atmosphere carbon

fluxes include prior and posterior estimates. The corresponding gridded uncertainties of NEE and ocean-atmosphere fluxes are also included in the flux files. The global gridded fluxes are 3-hourly with a resolution of 2° latitude × 2.5° longitude.

## 4 Results

### 4.1 Global carbon budget

Here, we present the five major components of the global carbon budget, including the fossil fuel $CO_2$ emissions ($E_{FOS}$),

biomass burning emissions ($E_{FIRE}$), atmospheric $CO_2$ concentration growth rate ($G_{ATM}$), ocean-atmosphere carbon fluxes ($F_{OCEAN}$), and NEE (Fig. 2). During the 2015–2022 period, $E_{FOS}$ was 9.71 ± 0.20 Pg C yr$^{-1}$, with a minimum of 9.44 Pg C yr$^{-1}$ in 2020 and a maximum of 9.94 Pg C yr$^{-1}$ in 2022; $E_{FIRE}$ was 1.86 ± 0.22 Pg C yr$^{-1}$, with a minimum of 1.47 Pg C yr$^{-1}$ in 2022 and a maximum of 2.14 Pg C yr$^{-1}$ in 2019. Over these 8 years, NEE exhibited a substantial mean sink with considerable interannual variability, estimated as the standard deviation across years (–4.08 ± 0.53 Pg C yr$^{-1}$). The sinks

were significantly weaker in 2015 and 2016 than in other years. The annual mean NEE in 2015–2016 was −3.35 Pg C yr$^{-1}$, whereas the NEE in 2017–2022 was −4.33 Pg C yr$^{-1}$. The reduced NEE during 2015-2016 was mainly related to the El Niño event, which caused substantial carbon release in the tropics (Wang et al., 2013; Liu et al., 2017; Piao et al., 2020; Dannenberg et al., 2021). Compared with NEE, interannual variation in the atmosphere-ocean fluxes was much smaller (−

2.32 ± 0.18 Pg C yr$^{-1}$). From 2015 to 2022, the terrestrial biosphere(NEE+$E_{\text{FIRE}}$)and ocean absorbed approximately 23%

and 24% of total fossil fuel $CO_2$ emissions, respectively, resulting in a $G_{\text{ATM}}$ of 5.17 ± 0.68 Pg C yr$^{-1}$.

We compared the GONGGA-estimated global carbon budget with results from the measurements and Global Carbon Budget 2023 (hereafter referred to as GCB2023) (Friedlingstein et al., 2023). The $G_{\text{ATM}}$ directly estimated from atmospheric $CO_2$ concentration measurements provided by the NOAA Earth System Research Laboratories Global Monitoring Laboratory (ESRL/GML) was 5.25 ± 0.61 Pg C yr$^{-1}$ during 2015–2023 (Lan et al., 2023), which corroborates the GONGGA

estimate. We also compared net biosphere exchange (NBE, i.e., the net carbon flux of all the land-atmosphere exchange processes except fossil fuel emissions, calculated as NEE+$E_{\text{FIRE}}$) and ocean sink estimated from the GONGGA inversion with GCB2023. Note that the GCB2023 estimations represent the carbon accumulated in the land and ocean reservoirs. We followed GCB2023's definitions and adjusted riverine $CO_2$ transport from the net atmosphere-surface $CO_2$ exchange over land (NBE) and ocean ($F_{\text{OCEAN}}$). Specifically, pre-industrial lateral carbon transport through the land-ocean aquatic

continuum (LOAC) of 0.65 ± 0.35 Pg C yr$^{-1}$ (Regnier et al. (2022) was subtracted from –NBE to represent land carbon sink, and added to –$F_{\text{OCEAN}}$ to represent ocean carbon sink. During 2015-2022, the mean of corrected land carbon sink from GONGGA was 1.57 ± 0.67 Pg C yr$^{-1}$, and the mean of corrected ocean sink was 2.97 ± 0.18 Pg C yr$^{-1}$. GCB2023's estimate of ocean sink was 2.88 ± 0.07 Pg C yr$^{-1}$ based on global ocean biogeochemistry models and surface ocean $f$CO$_2$-observation-based products. The land carbon sink from GCB2023 was 2.00 ± 0.62 Pg C yr$^{-1}$ from the dynamic global vegetation models

(DGVMs) and was 1.55 ± 0.77 Pg C yr$^{-1}$ calculated as the residual sink from the global budget of fossil fuel emissions, atmospheric growth rate and ocean sink (Friedlingstein et al., 2023). As the estimate of land carbon sink from DGVMs will introduce a budget imbalance in GCB2023, our estimates are well consistent with GCB2023's estimates based on ocean models and the residual land sink and close the global budget.

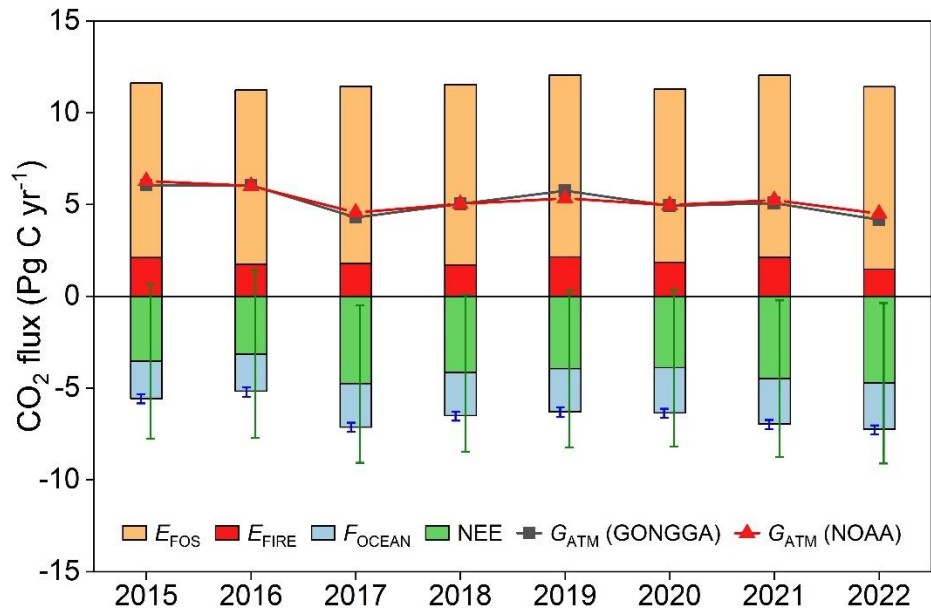

**Figure 2. Global carbon budget estimated by GONGGA and atmospheric CO₂ growth rate from NOAA during 2015–2022.**

## 4.2 Global distribution and regional fluxes

Figure 3 shows the global distributions of GONGGA annual mean NBE and ocean carbon fluxes during 2015–2022. Terrestrial carbon sinks were mainly in temperate North America, central South America, southern Africa, Europe, boreal Asia, India, eastern China, and most of Australia. Terrestrial carbon sources mainly occurred over western America, the eastern Amazon, central Africa, Southeast Asia, the southeastern coast of Australia, and New Zealand. The ocean sources mainly occurred over tropical oceans and the high-latitude Southern Ocean; the equatorial Pacific was the most prominent source area. Sinks mainly occurred over mid-latitude regions of both hemispheres and the high-latitude northern ocean. Generally, NBE had a more complex spatial distribution and higher uncertainty, compared with ocean carbon fluxes. Therefore, we explored the distribution and attribution of NBE over 11 TransCom land regions (Fig. 4) (Gurney et al., 2004).

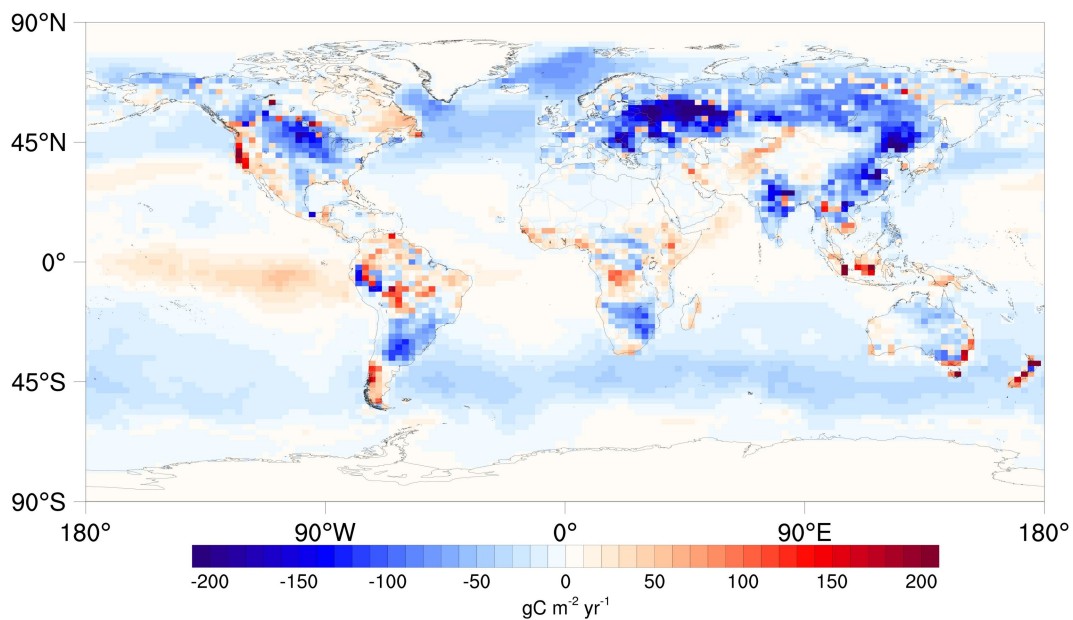

**Figure 3. GONGGA-estimated global distributions of annual mean (2015–2022) NBE and ocean carbon fluxes.**

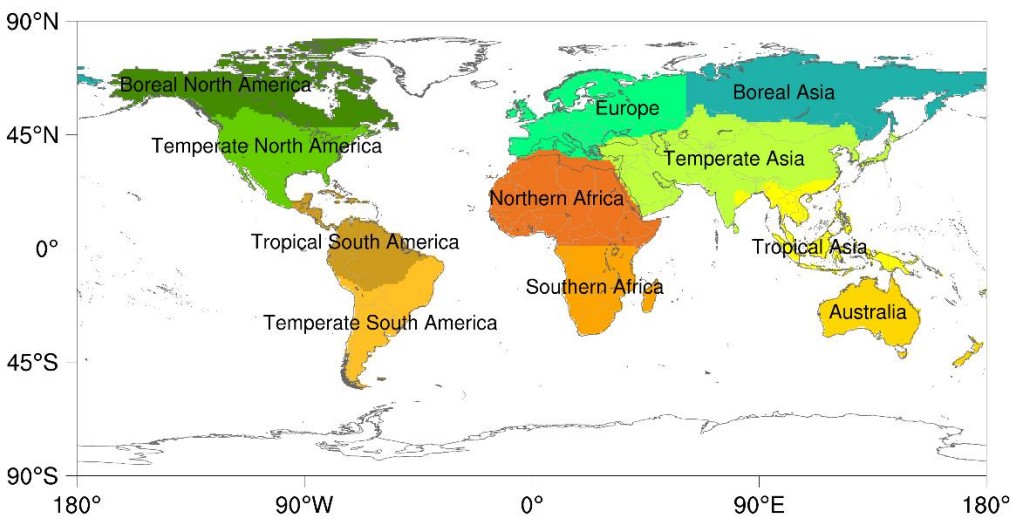

**Figure 4. Spatial distributions of 11 TransCom land regions.**

Here, we present the GONGGA-estimated annual mean (2015–2022) NBE for 11 TransCom land regions and their comparison with OCO-2 model intercomparison project (MIP) v10 inversions (Fig. 5). OCO-2 MIP v10 (Baker et al., 2023; Byrne et al., 2023) includes an ensemble of 14 atmospheric inversions over the period of 2015–2020 assimilating OCO-2 v10r retrievals, and each of them is characterized by distinct transport models, data assimilation algorithms, and prior fluxes (Table S1). We used OCO-2 MIP v10 results from the inversions that assimilate land nadir and land glint (LNLG) satellite

retrievals, and those assimilate in situ (IS) measurements. Here, in situ inversions are used to provide a baseline against satellite-driven results.

For the 11 TransCom regions, we estimated that Europe had the strongest terrestrial carbon sink, followed by Boreal Asia, Temperate Asia, Temperate North America, Temperate South America, Southern Africa, Boreal North America, and Australia, whereas Tropical South America, Northern Africa, and Tropical Asia were terrestrial carbon sources. All GONGGA and OCO-2 MIP LNLG and IS consistently indicated that Europe was the largest terrestrial sink. GONGGA showed good agreement with OCO-2 MIP inversions for most regions, and divergences occurred mainly in Boreal North America and Northern Africa. The difference between GONGGA and OCO-2 MIP inversions may be related to the prior NBE adopted and retrieval pre-processing methods utilized. In Boreal North America, GONGGA's prior emerged as a carbon source, whereas OCO-2 MIP's prior was a carbon sink (Fig. S1). After assimilating OCO-2 retrievals, GONGGA and OCO-2 MIP consistently showed Boreal North America was a carbon sink, but the sink in GONGGA was smaller than OCO-2 MIP.  The same situation happened in Northern Africa. Both GONGGA's prior and OCO-2 MIP's prior estimated Northern Africa as a terrestrial carbon sink, but the sink from GONGGA was stronger than that from OCO-2 MIP (Fig. S1). Constrained by OCO-2 retrievals, both GONGGA and OCO-2 MIP estimated it as a carbon source, and the source from GONGGA was weaker than that from OCO-2 MIP, aligning with the sizes of their prior sinks. In addition, the impact of prior fluxes may be amplified by the insufficient coverage of OCO-2 retrievals. For example, in Boreal North America, satellites cannot measure $XCO_2$ in dark high-latitude areas in winter. In Northern Africa, OCO-2 also has difficulties in accurately measuring $XCO_2$ over the desert because of its high albedo, demonstrated by its high proportion of bad retrievals (xco2_quality_flag = 1) (Zhang et al., 2016). Therefore, the posterior fluxes in these two regions were more dependent on the prior fluxes during the period with few OCO-2 retrievals. Notably, even in OCO-2 MIP inversions, the ensemble spread was prominent, indicating the difficulty of inversion in these regions using current satellite or in situ observations (Table S2).

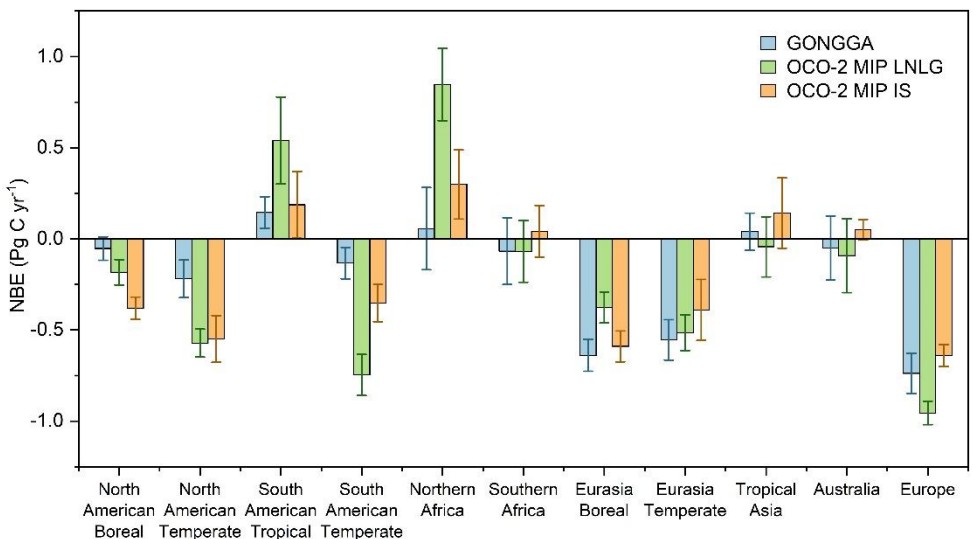

**Figure 5.** GONGGA annual mean (2015–2022) NBE for 11 TransCom land regions and comparisons with OCO-2 MIP LNLG and IS inversions. Error bars represent standard deviations in annual mean budget across the whole period.

### 4.3 Interannual variability and seasonal cycle

Here, we analyzed the interannual variability (IAV) and seasonal cycle of NBE at global and regional scales. We divided the globe into three large latitude bands: northern extratropics (30–90°N), tropics (30°S–30°N), and southern extratropics (90–30°S). The global net terrestrial carbon flux has a prominent year-to-year variability (Friedlingstein et al., 2022). We computed the standard deviation of global NBE to represent its magnitude of IAV, which amounted to 0.63 Pg C yr$^{-1}$ during the 2015–2022 period. The variations of NBE at northern extratropics, tropics, and southern extratropics were quite different (Fig. 6). We calculated the contribution of each latitude band to the global IAV using Eq. (1) from Ahlström et al. (2015). The contribution of the tropics to the global NBE IAV was 100.8%, whereas the contributions of the northern and southern extratropics were −13.2% and 12.4%, respectively. A positive (negative) score here indicates the variation is in the same (opposite) phase as the global IAV. The scores from our estimate indicated that the global IAV arises from the tropics. Given the short time series of the inversion, the latitudinal contributions in this study are suggestive but not statistically conclusive. The dominant role of tropical terrestrial ecosystems in the signal of the global carbon cycle IAV is consistent with previous results based on multiple observations and models (Baker et al., 2006; Rödenbeck et al., 2018b, a; Jung et al., 2020). Piao et al. (2020) reviewed and analyzed the regional contribution to global net terrestrial carbon flux IAV from 1980 to 2017 with process-based land carbon cycle models, atmospheric inversion models, and FLUXCOM data products. The contributions of the tropics to the global IAV obtained by these three methods were 83.4%, 71.7%, and 69.7%, respectively. In addition to the short time series, the inclusion of the 2015–2016 strong El Niño event in the period is an important reason for the large contribution score of the tropics in our estimate. Climatic variations are the main factors that drive the IAV of the net terrestrial carbon flux (Braswell et al., 1997; Zeng et al., 2005; Raupach et al., 2008; Liu et al., 2017). El Niño is the major climatic mode that modulates global temperature, precipitation, and solar radiation (Gu and Adler, 2011); thus, it drives the IAV of the carbon cycle (Bacastow, 1976; Rayner et al., 2008). The characteristics of hot and dry climate conditions in El Niño years are the primary reasons for the lower net carbon uptake or net carbon release by terrestrial ecosystems (Jones et al., 2001; Piao et al., 2009a), which is particularly evident in the tropics (Fig. S2) (Liu et al., 2017; Jin et al., 2023b). During 2015–2016, tropical land released 0.66 Pg C yr$^{-1}$ $CO_2$ into the atmosphere, whereas it is a net terrestrial sink in normal years (–0.52 Pg C yr$^{-1}$).

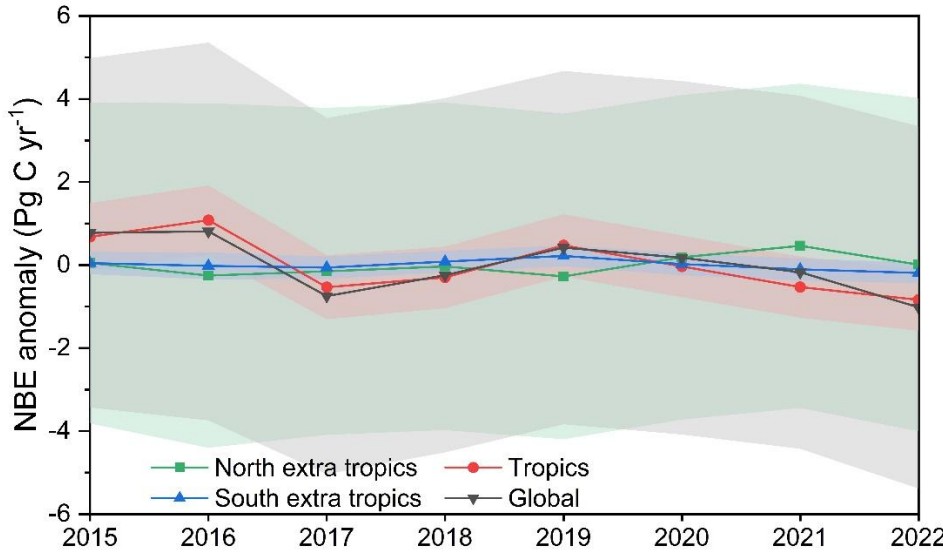

**Figure 6. Annual NBE anomaly over the globe, northern extra-tropics (30–90°N), tropics (30°S–30°N), and southern extra-tropics (90–30°S) during 2015–2022. The shadowed area represents the uncertainty of NBE in each region.**

The shape of the NBE seasonal cycle varies among regions and different years. In the northern extratropics, the size and phase of the seasonal cycle are very similar in all years, with July having the largest sink and northern winter being a carbon source. In the tropics, however, the seasonal cycles have smaller amplitudes and the shapes are distinct in different years. The largest deviations of the tropical seasonal cycle from the 8-year mean estimate are in 2016 ($R^2 = 0.34$, coefficient of determination between annual mean seasonal cycle and the year investigated) and 2019 ($R^2 = 0.50$); the most prominent deviations occurred during the peak 2015–2016 El Niño between July 2015 and June 2016 as well as 2019 El Niño between April 2019 and July 2019. The shape of the global seasonal cycle is nearly similar to the shape of the northern extratropics (with 103.6% contribution), whereas the tropics and southern extratropics have opposite phases compared with the global seasonal cycle (with −1.1% and −2.5% contribution, respectively). The dominance of the northern extratropics in the global seasonal cycle is consistent with previous findings (Forkel et al., 2016; Piao et al., 2020).

The amplitude is an important index of the seasonal cycle (i.e., seasonal cycle amplitude, SCA). The peak-to-trough amplitude was calculated as the difference between the maximum and minimum monthly NBE in each year. The 8-year mean SCA of NBE for the globe, northern extratropics, tropics, and southern extratropics were 3.55, 3.50, 0.43, and 0.12 PgC month$^{-1}$, respectively. The larger mean amplitude in northern land ecosystems, compared with other regions, was mainly related to the strong seasonality of gross primary production and ecosystem respiration (Randerson et al., 1997).

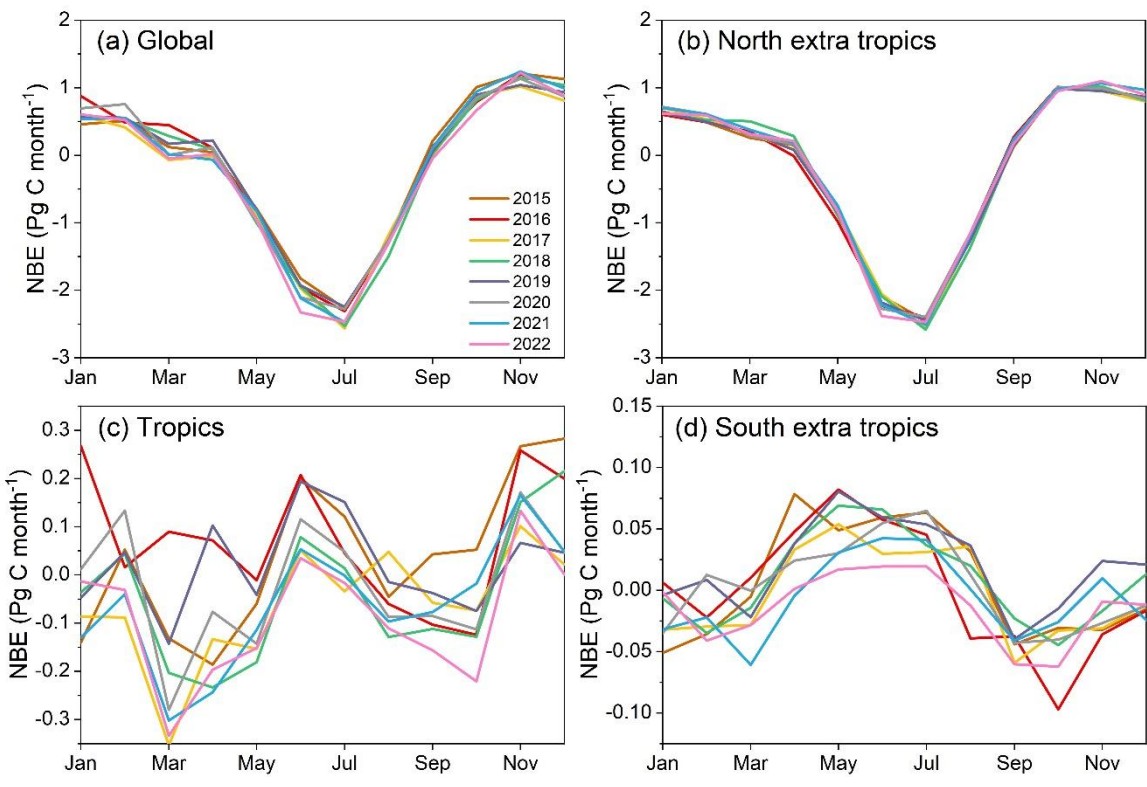

**Figure 7. Seasonal cycle of NBE for the globe, northern extratropics, tropics, and southern extratropics during 2015–2022.**

## 5 Dataset evaluation

### 5.1 Comparison with TCCON observations

In this section, we compare the simulated monthly mean $XCO_2$ driven by the posterior $CO_2$ fluxes with the retrievals from 27
TCCON sites during 2015–2022 (Table 1). The global mean root mean square error (RMSE) and bias between posterior simulated and TCCON retrieved $XCO_2$ were 0.81 and 0.24 ppm, respectively. Through the assimilation of OCO-2 retrievals, the atmospheric $CO_2$ simulations were considerably improved compared with prior simulations, which exhibited 1.15 ppm RMSE and 0.51 ppm bias at the global scale. At most sites, posterior RMSE was < 1 ppm, and bias was in the range of −0.5 to 1 ppm (Fig. 8). The maximum simulation deviation occurred at Eureka station (unless otherwise stated, "simulations"
hereafter refers to posterior simulations which means the simulation is driven by posterior fluxes), where an overestimation of simulated $XCO_2$ was observed in winter. This overestimation was also evident at Ny Ålesund and Sodankylä, which are located in the high latitudes of the Northern Hemisphere. Prior simulations generally overestimated $CO_2$ concentrations, particularly in winter (Fig. S3). Positive deviations were adequately mitigated at most sites after the inversion. However, for

the sites mentioned above, considering the lack of satellite retrievals in winter at high northern latitudes, the posterior flux
may be poorly constrained and is thus similar to the prior flux. Additionally, the coarse spatial resolution of the transport
model is another challenge for the detection of sub-grid variations in $XCO_2$. For example, Edwards station and Pasadena
station are close to each other, and they are located in the same grid cell of the transport model. The simulated $XCO_2$ time
series at these two sites are similar, and the minor difference mainly arises from the interpolation process (Fig. S4). In
contrast, the TCCON $XCO_2$ retrievals are considerably higher at Pasadena station than at Edwards station, with a multi-year
mean difference of 0.84 ppm.

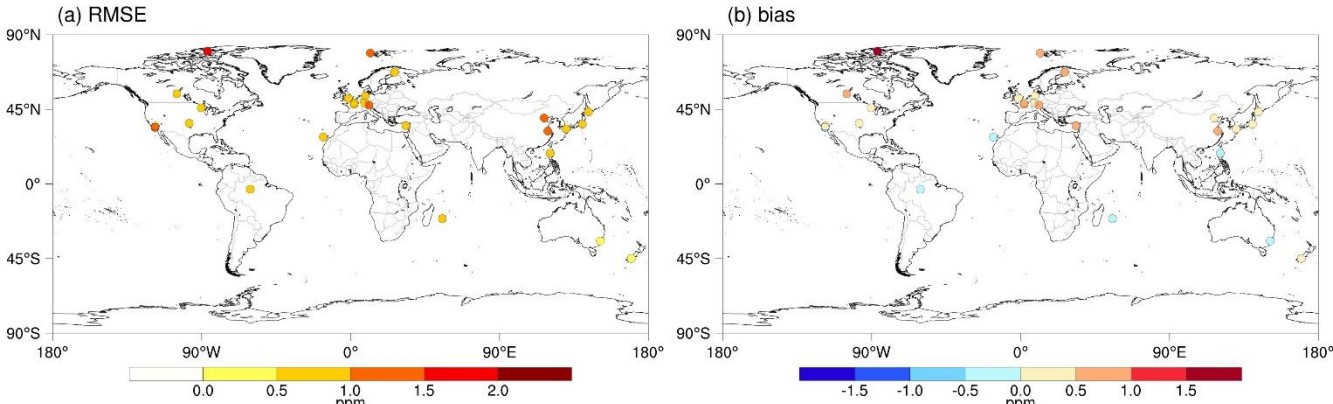

**Figure 8. Spatial distributions of (a) root mean square error (RMSE) and (b) bias between the posterior monthly $XCO_2$ simulations and corresponding observations at each TCCON site (simulations minus observations; unit: ppm).**

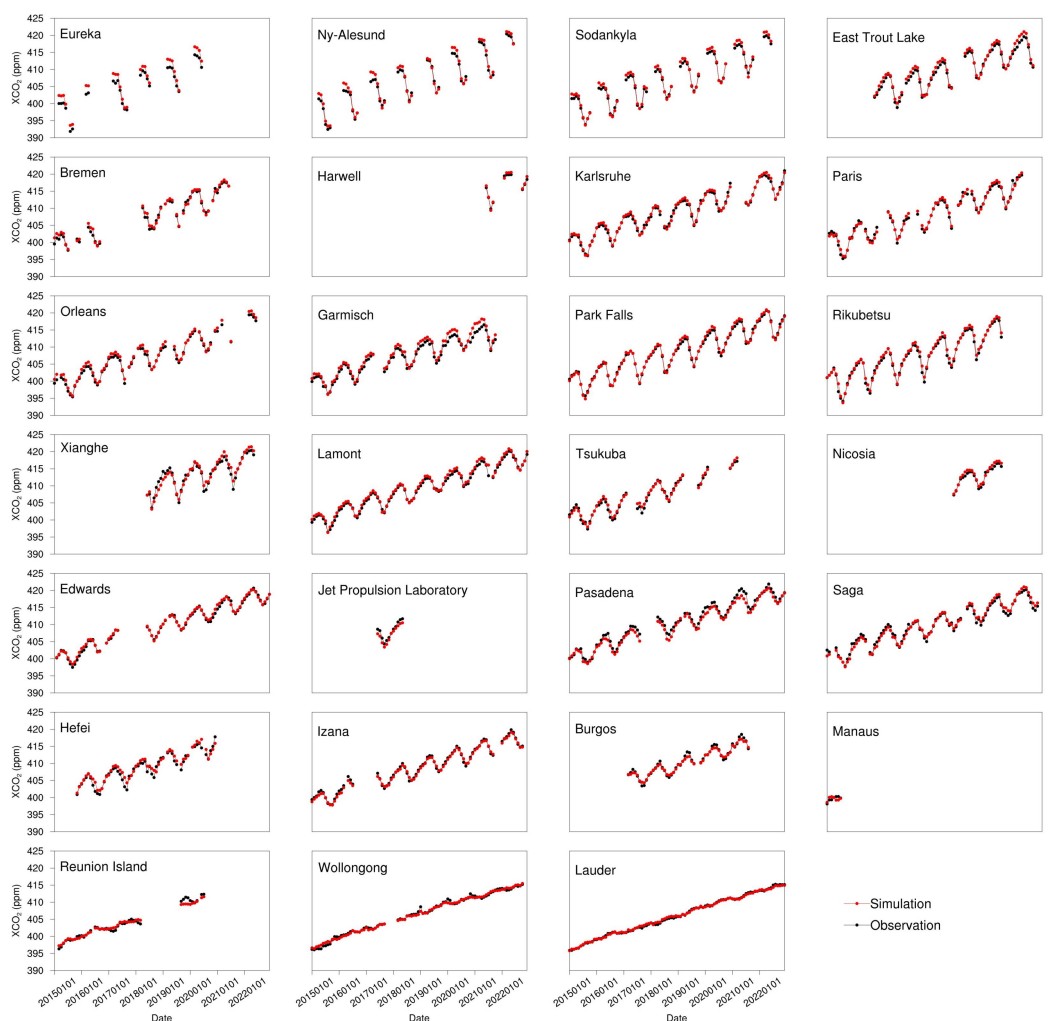

**Figure 9. Time series of monthly averaged observations and posterior simulations at each TCCON site.**

### 5.2 Comparison with ObsPack observations

Here, we compare posterior $CO_2$ simulations with ObsPack surface flask and aircraft observations. The global mean RMSE

and bias between surface flask observations and corresponding simulations were 1.76 and −0.33 ppm, respectively. For most surface flask sites located on the ocean and in tropical and southern extratropical terrestrial regions, RMSE was < 2.0 ppm; bias was in the range of −0.5 ppm to 0.5 ppm. The high model–data RMSE values mainly occurred over northern middle latitudes, particularly over Europe and East Asia. Jiang et al. (2022) used GOSAT $XCO_2$ retrievals to estimate global $CO_2$ fluxes and also found that posterior $CO_2$ concentrations could differ from surface observations, mainly in the northern

extratropics. Because of limitations regarding the coarse resolutions of global transport models and thus differences in representativeness between simulated $CO_2$ concentrations and actual observations over land, some sites have significant

data–model mismatches. For example, at the three sites with posterior RMSE values exceeding 4.0 ppm, the observed atmospheric $CO_2$ concentrations had strong temporal fluctuations, which were presumably caused by localized and short-term surface fluxes (Fig. S5).

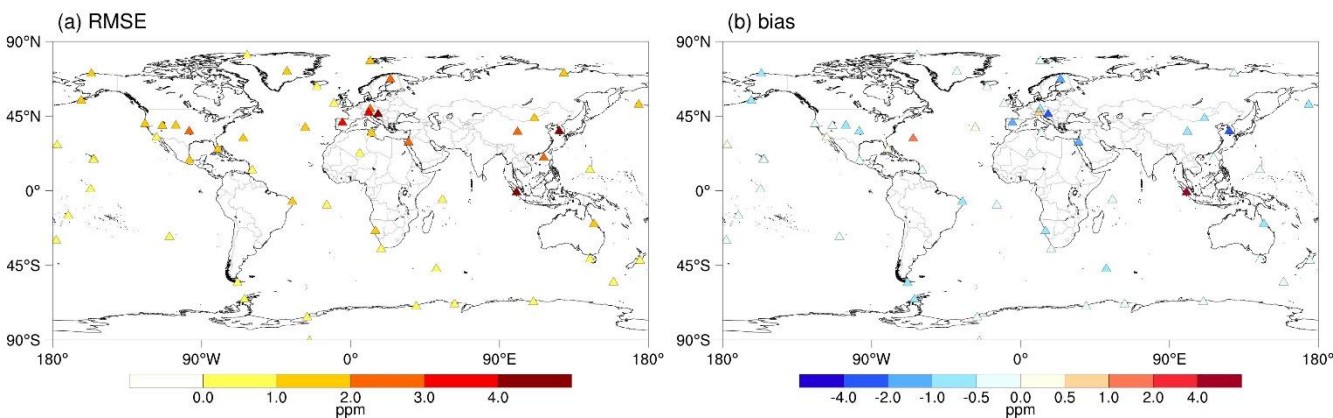

**Figure 10. Spatial distributions of (a) RMSE and (b) bias between the posterior monthly $XCO_2$ simulations and corresponding observations at each surface flask site (posterior simulations minus observations; unit: ppm).**

To decrease mismatches in temporal and spatial representativeness between observations and simulations, we compared the monthly observed and simulated $CO_2$ concentrations in six land regions (Fig. 11). Apart from RMSE and bias, we further present the random error here, which was calculated as the standard deviation of the differences between simulated and observed $CO_2$ concentrations (Rastogi et al., 2021). The monthly simulations closely agreed with the observations. Bias was in the range of −0.44 to −1.27 ppm, random error was in the range of 0.39 to 1.65 ppm, and RMSE was in the range of 0.58 to 2.08 ppm. The simulation deviations remained higher for North America, Europe, and East Asia, compared with other regions. In these three regions, there was a significant difference in terms of comparisons with TCCON and ObsPack surface flask observations. Mainly positive bias arose from TCCON evaluations and negative bias arose from ObsPack evaluations. This discrepancy may be related to the nature of the two types of observations. TCCON observations are column-averaged atmospheric $CO_2$ concentrations, whereas ObsPack observations are surface atmospheric $CO_2$ concentrations. The opposite signs of bias between the two comparisons may be related to the imperfect simulation of vertical mixing of GEOS-Chem (Schuh et al., 2019).

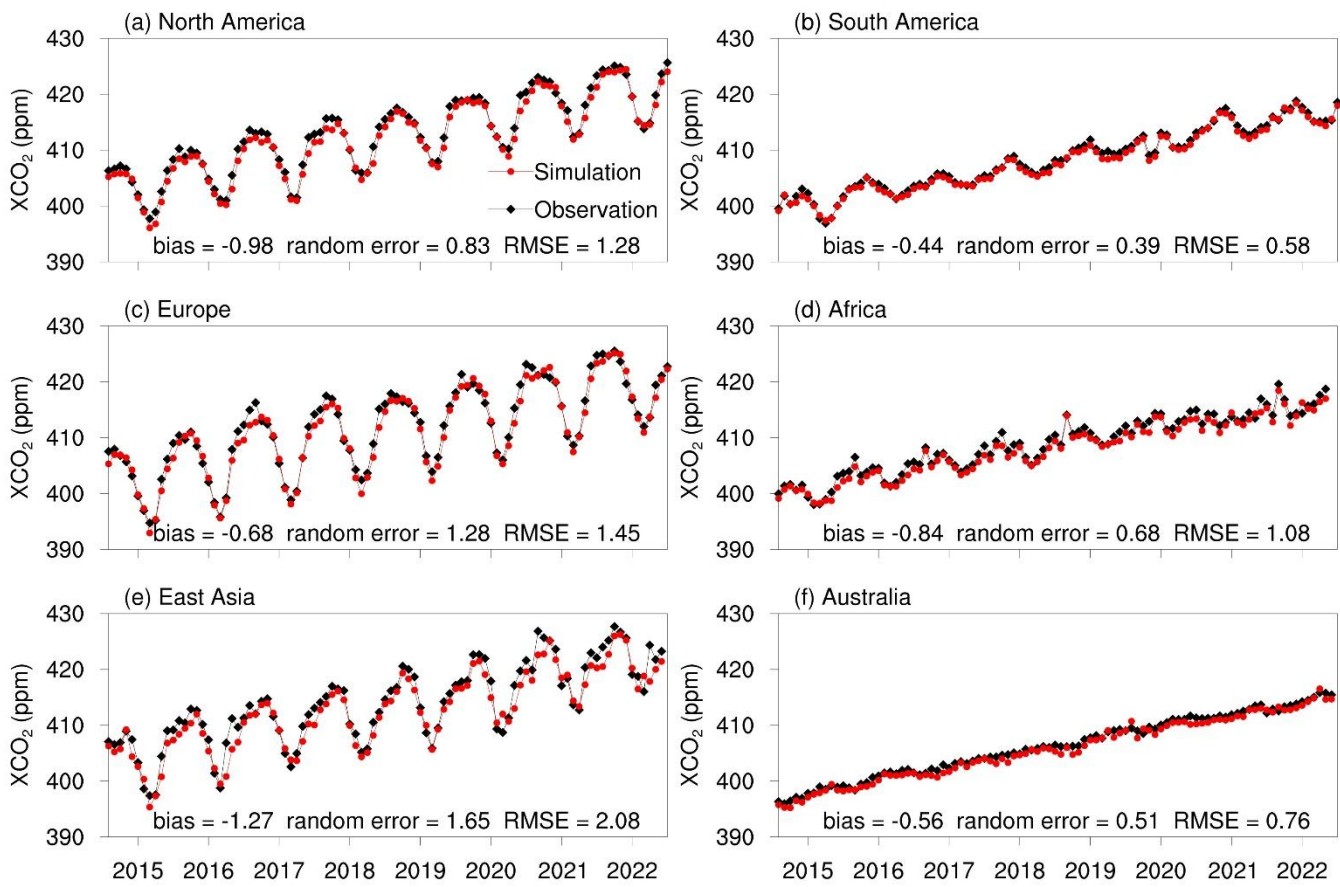

**Figure 11. Time series of monthly averaged ObsPack surface flask observations and corresponding posterior simulations for the six sub-regions.**

For aircraft observations, we calculated the mean statistics of each grid cell (Fig. 12). The simulations closely agreed with the aircraft observations. For most grid cells, the RMSE was < 2.0 ppm; bias was between −1.0 and 1.0 ppm. The simulated deviations over Boreal North America and Temperate North America were generally larger than over the ocean, similar to the surface flask results. We also compared the vertical distribution of modelled $CO_2$ against the observations. Figure S6 shows that the random errors were typically within 1 ppm, showing a good agreement between the simulations and observations. However, large biases, up to 2 ppm, were seen in the high latitudes and above 9 km, consistent with the comparisons against the TCCON retrievals (Fig. 8).

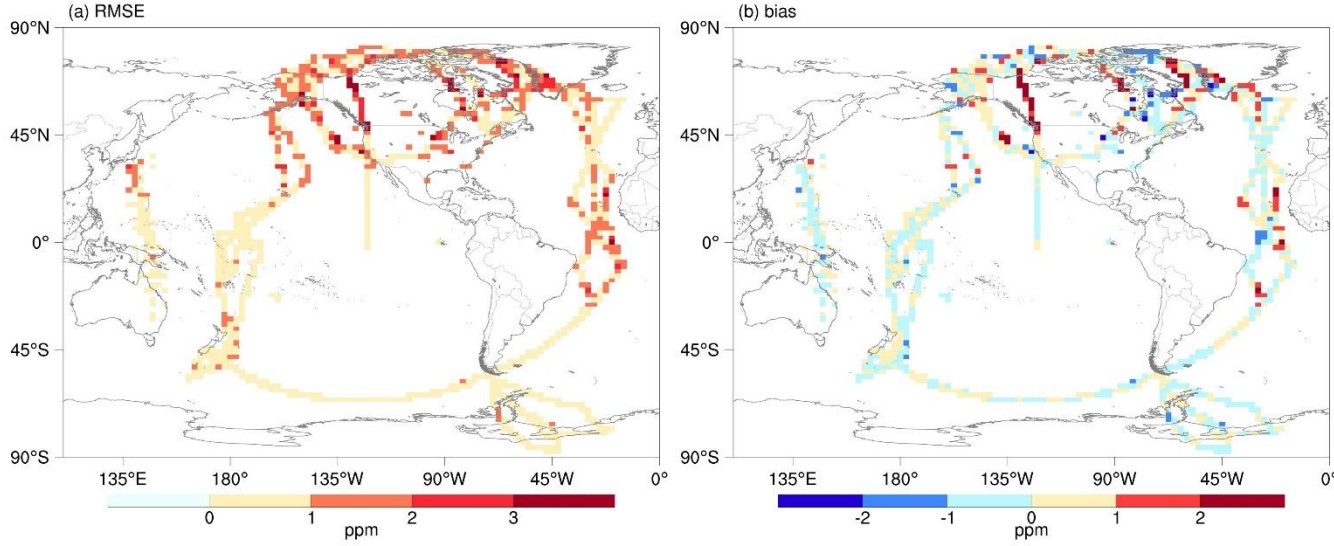

**Figure 12. The (a) random error and (b) bias between posterior CO$_2$ simulations and aircraft observations at each grid cell (posterior simulations minus observations; unit: ppm).**

## 6 Discussion

Regarding the regional carbon budget, we found that fire emission, although it was not optimized in the inversion, largely impacted the net CO$_2$ fluxes from terrestrial ecosystem, i.e. NBE, in equatorial regions and Australia. With the frequent occurrence of wildfires in these regions, carbon emissions from wildfires may exceed regional NEE and make these regions net carbon sources (Fig. S7). For the past few decades, ~50% of fire-related carbon emissions and ~70% of global burned areas occurred across African subtropical savannah systems (Giglio et al., 2013; Andela and Van Der Werf, 2014). In the Amazon, despite the decline in deforestation rate during 2003-2015, carbon emissions from drought-induced fires had increased very quickly (Aragão et al., 2018). Southeast Australia also experienced intensive and geographically extensive wildfires during the 2019–2020 summer season, and the fires released substantial amounts of CO$_2$ into the atmosphere (Wang et al., 2020a; Byrne et al., 2021; Van Der Velde et al., 2021). As a result, the 8-yr mean biomass burning emissions in Tropical South America, Northern Africa, and Tropical Asia amounted to 0.17, 0.33, and 0.13 Pg C yr$^{-1}$, and were 6.2, 1.2, and 1.4 times higher than regional NEE, respectively, resulting in net carbon sources in these regions. The increasing fire emissions thus present a great challenge to climate mitigation efforts.

The processing of XCO$_2$ uncertainties also had an impact on the inversion results. We performed three sensitivity inversions with different XCO$_2$ uncertainties. The XCO$_2$ uncertainties were inflated two and four times in the first (E1) and second (E2) test, respectively. In the third test (E3), the XCO$_2$ uncertainties were increased by 5 ppm. The three sensitivity tests adopted the same configuration as the reference inversion in this study only except for the XCO$_2$ uncertainties. The distributions of different XCO$_2$ uncertainties were shown in Fig. S8. At the global scale, the inverted annual NBE and

$F_{OCEAN}$ from the original inversion, E1, and E2 were very close, but E3 had a different partitioning between land and ocean fluxes than the other inversions, which amounted to about 0.2 Pg C yr$^{-1}$ (Fig. S9). When it comes to regional scale, the differences were larger in some regions and years but were still broadly consistent with the reference inversion (Fig. S10). These sensitivity tests highlighted the fact that the inversion results were indeed impacted by the assumption regarding $XCO_2$ uncertainty and careful assessment of uncertainties in satellite $XCO_2$ retrievals was necessary for accurate estimates of global and regional carbon fluxes.

In the current version of GONGGA, we assimilated the OCO-2 v11r Lite $XCO_2$ dataset. A recent paper found that the v11r Lite product has a bias of -0.4 to -0.8 ppm across regions north of 60°N due to the variations of digital elevation model (DEM) used in the retrieval algorithm (Jacobs et al., 2024), and this bias introduces a ~ 100 Tg C shift in the partitioning of carbon fluxes for the latitudinal bands. A preliminary test of GONGGA using the latest v11.1r Lite product showed the inverted terrestrial carbon sink tends to be 90 to 140 Tg C yr$^{-1}$ lower north of 60° N than using the v11r Lite product, consistent with the previous findings. In addition, some parts of GONGGA's inversion algorithm, such as the data selection, were partly different from those proposed by the OCO-2 Science Team (Baker et al., 2022; Peiro et al., 2022; Byrne et al., 2023), but GONGGA's inversion results were broadly consistent with the ensemble of OCO-2 MIP inversions and GCB2023, and gave reasonable estimates of global and regional carbon budgets within the uncertainties. In the future, GONGGA will regularly publish new versions of inverted fluxes using the latest OCO-2 data on an annual basis. These updates will align with the latest suggestions from the OCO-2 Science Team, enabling the ongoing monitoring of $CO_2$ fluxes.

## 7 Data availability

The dataset is available at https://doi.org/10.5281/zenodo.8368846 (Jin et al., 2023a). As the satellite $XCO_2$ retrievals, prior carbon fluxes, and meteorological data are persistently improving and updating, we plan to update the dataset annually in the future, aiming to support scientific research and policy making.

## 8 Summary

Here, we presented a global resolved surface carbon flux dataset during the 2015–2022 period. The dataset includes 3-hourly gridded (2° latitude × 2.5° longitude) NEE and ocean carbon fluxes (prior and posterior), together with prescribed fossil fuel emissions and biomass burning emissions. The dataset was generated by the GONGGA inversion system constrained by OCO-2 $XCO_2$ retrievals. We analyzed the key characteristics of the global and regional carbon cycles in terms of carbon budget, interannual variability, and seasonal cycle. The global annual estimate from GONGGA was consistent with the estimate from GCB2022. Regional fluxes were analyzed based on TransCom partitions. The strongest carbon sinks were observed in Europe, followed by Boreal Asia and Temperate Asia. We validated posterior fluxes by comparing posterior

simulated $CO_2$ concentrations with TCCON $XCO_2$ retrievals, as well as ObsPack surface flask and aircraft observations. Both evaluations demonstrated the optimization of posterior fluxes through assimilation of OCO-2 satellite retrievals.

**Author contributions.** TX conceptualized, administrated, and supervised the research, and acquired funds for it. JZ and WY made investigations, developed the inversion system, and visualized the data. JZ created the dataset. JZ, ZH and ZM made formal analysis of it. TX and ZH developed the methodology. TX and PS provided necessary resources. JZ wrote the original manuscript draft. TX, WY, WT, DJ and PS reviewed and edited the manuscript draft.

**Competing interest.** The authors declare that they have no conflict of interest.

**Acknowledgements.** The OCO-2 data are provided by the ACOS/OCO-2 project at the Jet Propulsion Laboratory, California Institute of Technology, and can be obtained from the data archive at the NASA Goddard Earth Science Data and Information Services Center. We acknowledge all atmospheric data providers for obspack_co2_1_GLOBALVIEWplus_v8.0_2022-08-27 and obspack_co2_1_NRT_v8.1_2023-02-08. We acknowledge the TCCON science team, and the TCCON data were obtained from the TCCON Data Archive hosted by CaltechDATA at https://tccondata.org. .

**Financial support.** This work was supported by the Second Tibetan Plateau Scientific Expedition and Research Program (2022QZKK0101) and the National Natural Science Foundation of China (Grant Nos. 41988101, 41975140, 42001104), the Innovation Program for Young Scholars of TPESER (TPESER-QNCX2022ZD-01).

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
