# Peer review of "A global surface CO2 flux dataset (2015–2022) inferred from OCO-2 retrievals using the GONGGA inversion system"

_Earth System Science Data, 2023_

## Author Comment (AC1)

**A global surface $CO_2$ flux dataset (2015–2022) inferred from OCO-2 retrievals using the GONGGA inversion system**
**Manuscript No. essd-2023-449**

Zhe Jin, Xiangjun Tian, Yilong Wang, Hongqin Zhang, Min Zhao, Tao Wang, Jinzhi Ding, Shilong Piao

**Reply to reviewer 1**

We would like to thank reviewer 1 for this thorough review and insightful comments, which significantly improved this manuscript. Please see below the point-to-point responses. All the line numbers indicated in this letter of response correspond to the revised version of text.

**Reviewer 1:**

*This manuscript presents an 8-year dataset of surface-atmosphere $CO_2$ fluxes estimated by the GONGGA inversion system constrained by OCO-2 $XCO_2$ retrievals. This provides a useful dataset to the community and the paper is well written and structured to present the dataset and its evaluation. However, I feel that some important details are missing as described below. I recommend publication after addressing the following minor comments.*

**Response:** We thank the reviewer for the positive evaluation of our study and the valuable suggestions to improve it. We have carefully revised our manuscript following the comments and suggestions.

Main comments:
*1.   Sec. 2.1: Some details are missing here. What is the spatial and temporal resolution of the optimization. Is it at 2x2.5 and monthly (weekly?)? How is the covariance between surface flux and atmospheric $CO_2$ constructed? If the set-up follows the set-up of a previous study that explicitly state this?*
**Response:**

We optimized the surface flux at each grid cell of the transport model GEOS-Chem, where the size of each grid cell is $2° \times 2.5°$ (latitude×longitude). The temporal resolution of the optimization is 14 days, the same as the length of an inversion window.

GONGGA adopted a dual-pass inversion strategy, which first optimized the initial $CO_2$ concentrations in the $CO_2$ pass, and then optimized surface $CO_2$ fluxes in the flux pass (Jin et al., 2023). In the $CO_2$ pass, the prior error covariance was constructed through sampling from a historical $CO_2$ simulation. In the flux pass, the prior error covariance was constructed through the primary modes of historical fluxes, which would be introduced in more detail in the response to the second question. As these two

passes were performed successively, the prior errors from $CO_2$ concentrations and fluxes were assumed to be independent.

The link between surface flux and atmospheric $CO_2$ is constructed through the observation operator $H(\cdot)$ which relies on GEOS-Chem simulations and sampling of modelled atmospheric $CO_2$. Specifically, the atmospheric transport model is first used to simulate gridded $CO_2$ concentrations driven by surface fluxes. Then, the simulated gridded $CO_2$ profiles are interpolated horizontally by inverse distance weighting and vertically by linear interpolation on pressure. At last, the interpolated $CO_2$ profiles are used to construct the simulated $XCO_2$ with the equation:

$$XCO_2^m = XCO_2^a + \mathbf{h}^{\mathrm{T}}\mathbf{A}(\mathbf{x}_{CO2} - \mathbf{x}_{CO2,a}).$$

where $XCO_2^m$ is the modelled $XCO_2$, $XCO_2^a$ is the prior value provided by the OCO-2 Lite file, $\mathbf{h}$ is the pressure weighting function, $\mathbf{A}$ is the averaging kernel matrix, $\mathbf{x}_{CO2}$ is the interpolated $CO_2$ profile, and $\mathbf{x}_{CO2,a}$ is the prior $CO_2$ profile provided by the OCO-2 Lite file.

The set-up of GONGGA has been described in the previous study (Jin et al., 2023). In the revised manuscript, we added the description of the optimized fluxes and how they are connected with atmospheric $CO_2$ data:

- Line 122-124: "The spatial resolutions of the optimization for both initial $CO_2$ concentrations and fluxes are 2° latitude × 2.5° longitude, the same as the transport model resolution. The temporal resolution of the optimization is 14 days, indicating that the fluxes within each 14-day window are uniformly adjusted by the same scaling factor."

- Line 107-115: "$H(\cdot)$ is the observation operator, which relies on GEOS-Chem simulations and sampling of modelled atmospheric $CO_2$. Firstly, the atmospheric transport model is used to simulate gridded $CO_2$ concentrations driven by surface fluxes. Then, the simulated gridded $CO_2$ profiles are interpolated horizontally by inverse distance weighting and vertically by linear interpolation on pressure. Thirdly, the interpolated $CO_2$ profiles are used to construct the simulated $XCO_2$ using the equation:

$$XCO_2^m = XCO_2^a + \mathbf{h}^{\mathrm{T}}\mathbf{A}(\mathbf{x}_{CO2} - \mathbf{x}_{CO2,a}). \tag{2}$$

where $XCO_2^m$ is the modelled $XCO_2$, $\mathbf{x}_{CO2}$ is the interpolated $CO_2$ profile from the GEOS-Chem simulation. $XCO_2^a$, $\mathbf{h}$, $\mathbf{A}$, and $\mathbf{x}_{CO2,a}$ are the prior value of $XCO_2$, the pressure weighting function, the averaging kernel matrix, and the prior $CO_2$ vertical profile, respectively, provided by the OCO-2 Lite file."

*2. Sec. 2.2: How was prior error covariance matrix created? Is it diagonal? Is it an output of ORCHIDEE-MICT? Same question for ocean flux uncertainties. Based of Fig. 2 it seems that the global land and ocean uncertainties are very different in magnitude, despite the fact that the GCP gives similar order of magnitude uncertainties, why is this?*
**Response:**
The prior error covariance matrix was built using the ensemble perturbations (Text S1). It accounts for the spatial error covariances between fluxes at different grid cells

in the off-diagonal elements. Specifically, the prior perturbations of the scaling factors in the first inversion window were obtained through historical sampling. We first created a large ensemble of 108 samples of gridded monthly mean fluxes from January 2011 to December 2019, and divided them by the gridded fluxes of the first month of the inversion, with a value of 1 subsequently subtracted to form the ensemble of prior perturbations of scaling factors. Then we extracted 36 samples that could represent the primary modes of the big sample using Random State Variable (RSV) method (Zhang et al., 2020). Each sample comprises 91×144 (latitude×longitude) perturbation values, corresponding to one value per grid cell. The prior perturbations in the subsequent inversion windows were updated through the inversion following the method described in Tian et al. (2020). Both NEE and ocean-atmosphere fluxes applied this sample generation method. The historical NEE were from ORHIDEE-MICT simulations (Guimberteau et al., 2018), and historical ocean-atmosphere fluxes were from Takahashi climatology results (Takahashi et al., 2009). In the revised manuscript, we added the description on how the **B** matrix was built in Text S1:

- "The prior perturbations of the scaling factors in the first inversion window were obtained through historical sampling of fluxes. We first created 108 samples from historical fluxes, which consists of the monthly mean fluxes from January 1, 2011 to December 31, 2019. Then they divided the monthly mean flux in September 2014 and subtracted 1 to form the ensemble of perturbations of flux scaling factors. Subsequently, 36 samples that could represent the key spatial patterns of the large ensemble were extracted using Random State Variable (RSV) method (Zhang et al., 2020), forming the prior perturbations for the first inversion window. After the inversion of the first window, the prior perturbations of the next window were updated (Tian et al., 2020):

$$\mathbf{P}_x^{prior,w+1} = \mathbf{P}_x^{prior,w} \mathbf{V}_2 \Phi^{\mathrm{T}} \tag{S3}$$

Where $\mathbf{P}_x^{prior,w+1}$ is the ensemble of the prior perturbations for the next window, and $\mathbf{P}_x^{prior,w}$ is the ensemble of the prior perturbations for the current window. The matrix $\mathbf{V}_2$ can be calculated by Eq. (S5-S7) detailed below, and $\Phi^{\mathrm{T}}$ is a random orthogonal matrix. The procedure was repeated through all inversion windows. Both NEE and ocean-atmosphere fluxes applied this sample generation method. The historical NEE were from ORHIDEE-MICT simulations (Guimberteau et al., 2018), and historical ocean-atmosphere fluxes were from Takahashi climatology results (Takahashi et al., 2009). As a result, the total uncertainty of our prior land and ocean fluxes at a global scale and for a full year, before assimilating $XCO_2$ observations, amount to an average of 4.7 Pg C yr$^{-1}$ and 0.28 Pg C yr$^{-1}$, respectively."

Figure 2 shows the posterior uncertainty of land and ocean fluxes, which are the estimates from the Bayesian statistics. They are different from the large spread of the ensemble of process-based models or inversion models in GCP, which encompasses many more sources of uncertainties. The variations among process-based models come from different processes included, different equations to describe the key processes, and their parameterizations. The inversion models can vary in the atmospheric transport

models, the inversion algorithm, the prior fluxes, and assimilated observations. We think it is difficult to directly compare the Bayesian statistic posterior uncertainty with the spread of GCP model ensemble. In GONGGA, the total annual uncertainty of our prior land and ocean fluxes on a global scale, before assimilating $XCO_2$ observations, amount to 4.7 Pg C $yr^{-1}$ and 0.28 Pg C $yr^{-1}$, which is on the same order of magnitude as other inversion systems. For instance, the 1-σ uncertainty for the prior land and air-sea fluxes in the CAMS inversion system (Chevallier, 2021) are 3.0 Pg C $yr^{-1}$ and 0.5 Pg C $yr^{-1}$, respectively.

For the posterior uncertainty shown in Fig. 2, we found a code mistake in our calculation, which did not account for the error coherence in time within an inversion window as the 14 days share the same scaling factor. After the correction, the average posterior uncertainty of land and ocean fluxes are 4.3 Pg C $yr^{-1}$ and 0.25 Pg C $yr^{-1}$, respectively. We updated the posterior uncertainty in Fig. 2 in the revised manuscript, which is also shown here as Fig. R1.

[Figure]

**Figure R1. Global carbon budget estimated by GONGGA and atmospheric $CO_2$ growth rate from NOAA during 2015–2022.**

*3.     Sec 2.3. The OCO-2 $XCO_2$ dataset is not properly cited. There is the v11r standard $XCO_2$ product (no bias correction, JPL DEM, still running routinely), the v11r Lite $XCO_2$ product (bias corrected, JPL NASADEM+, available up to April 2023) and the v11.1r Lite $XCO_2$ product (bias corrected, Copernicus DEM, still running routinely). Please clearly state and cite which dataset was used. An important point is that the DEM used in v11r cause a systematic error over the northern high latitudes that may have impacted the inversion results, if used. The impact of the DEM change is described in Jacobs et al. (2023): https://doi.org/10.5194/amt-2023-151. Instructions for citing the OCO-2 retrievals are given on the GES DISC website. For example, if this was V11.1r downloaded from GES DISC then citation should be: OCO-2/OCO-3 Science Team, Vivienne Payne, Abhishek Chatterjee (2022), OCO-2 Level 2 bias-corrected*

*XCO2 and other select fields from the full-physics retrieval aggregated as daily files, Retrospective processing V11.1r, Greenbelt, MD, USA, Goddard Earth Sciences Data and Information Services Center (GES DISC), Accessed: **[Data Access Date]**, 10.5067/8E4VLCK16O6Q"*

**Response:**

We used the v11r Lite $XCO_2$ product in this submission. We thank the reviewer for directing us to the recent paper on the potential biases in v11r $XCO_2$ retrievals in the northern high latitudes. We are now starting to run the inversion with the latest v11.1r product. The preliminary results from 2015 to 2018 show that the inverted land carbon sinks north of 60°N using v11.1r are smaller than those using v11r by 90 to 140 Tg C $yr^{-1}$, and accompanied by a compensating increase in ocean carbon uptake in the northern mid- and low- latitudinal band (Fig. R2).

In the revised manuscript, we added the reference for the data version and a paragraph to discuss the uncertainty related to the different versions of OCO-2 retrieval products:

- Line 417-427: "In the current version of GONGGA, we assimilated the OCO-2 v11r Lite $XCO_2$ dataset. A recent paper found that the v11r Lite product has a bias of -0.4 to -0.8 ppm across regions north of 60°N due to the variations of digital elevation model (DEM) used in the retrieval algorithm (Jacobs et al., 2024), and this bias introduces a ~ 100 Tg C shift in the partitioning of carbon fluxes for the latitudinal bands. A preliminary test of GONGGA using the latest v11.1r Lite product showed the inverted terrestrial carbon sink tends to be 90 to 140 Tg C $yr^{-1}$ lower north of 60° N than using the v11r Lite product, consistent with the previous findings. In addition, some parts of GONGGA's inversion algorithm, such as the data selection, were partly different from those proposed by the OCO-2 Science Team (Peiro et al., 2022; Byrne et al., 2023; Baker et al., 2022), but GONGGA's inversion results were broadly consistent with the ensemble of OCO-2 MIP inversions and GCB2023, and gave reasonable estimates of global and regional carbon budgets within the uncertainties. In the future, GONGGA will regularly publish new versions of inverted fluxes using the latest OCO-2 data on an annual basis. These updates will align with the latest suggestions from the OCO-2 Science Team, enabling the ongoing monitoring of $CO_2$ fluxes."

[Figure]

**Figure R2. The inverted (a) NBE and (b) $F_{OCEAN}$ in regions north of 60°N and 0-60°N from GONGGA inversions using OCO-2 v11r retrievals (orange bars) and v11.1r (green bars) retrievals during 2015-2018.**

*4.   L190-191: I think that the definition "$S_{LAND}$" is confusing here. In the Global Carbon Budget papers, the term $S_{LAND}$ is the net land sink after accounting for net land-use change emissions. However, in this paper, $S_{LAND}$ is defined as NEE (e.g., $S_{LAND}$ = NBE – Fire). But fire does not equal $E_{LUC}$, so the definitions are different. I recommend not using $S_{LAND}$ to define this quantity. It may be best to compare the NBE terms between the two studies after accounting for lateral fluxes. I recommend reviewing Sec. 7 of Byrne et al. (2023; https://essd.copernicus.org/articles/15/963/2023/) to see a comparison between the OCO-2 v10 MIP and Global Carbon Budget numbers.*
**Response:**
      Thank you for clarifying the differences between the two flux terms. To avoid confusion, we changed the notation and reported NEE for the terrestrial atmosphere-surface $CO_2$ exchange except biomass burning emissions, which is the value directly estimated by the inversion. In addition, when comparing our inversion results with GCB estimates in the last paragraph of Section 4.1, we adjusted GONGGA's NBE estimates to account for the lateral flux of carbon transported by rivers as reported by GCB. Note that we also changed GCB values to the latest estimates (GCB2023) in the revised manuscript. The manuscript was revised as follows:
-     Line 203-205: "Here, we present the five major components of the global carbon budget, including the fossil fuel $CO_2$ emissions ($E_{FOS}$), biomass burning emissions ($E_{FIRE}$), atmospheric $CO_2$ concentration growth rate ($G_{ATM}$), ocean $CO_2$ flux ($F_{OCEAN}$), and NEE (Fig. 2)."

- Line 219-232: "We also compared net biosphere exchange (NBE, i.e., the net carbon flux of all the land-atmosphere exchange processes except fossil fuel emissions, calculated as NEE+$E_{FIRE}$) and ocean sink estimated from the GONGGA inversion with GCB2023. Note that the GCB2023 estimations represent the carbon accumulated in the land and ocean reservoirs. We followed GCB2023's definitions and adjusted riverine $CO_2$ transport from the net atmosphere-surface $CO_2$ exchange over land (NBE) and ocean ($F_{OCEAN}$). Specifically, pre-industrial lateral carbon transport through the land-ocean aquatic continuum (LOAC) of $0.65 \pm 0.35$ Pg C yr$^{-1}$ (Regnier et al. (2022) was subtracted from –NBE to represent land carbon sink, and added to –$F_{OCEAN}$ to represent ocean carbon sink. During 2015-2022, the mean of corrected land carbon sink from GONGGA was $1.57 \pm 0.67$ Pg C yr$^{-1}$, and the mean of corrected ocean sink was $2.97 \pm 0.18$ Pg C yr$^{-1}$. GCB2023's estimate of ocean sink was $2.88 \pm 0.07$ Pg C yr$^{-1}$ based on global ocean biogeochemistry models and surface ocean $fCO_2$-observation-based products. The land carbon sink from GCB2023 was $2.00 \pm 0.62$ Pg C yr$^{-1}$ from the dynamic global vegetation models (DGVMs) and was $1.55 \pm 0.77$ Pg C yr$^{-1}$ calculated as the residual sink from the global budget of fossil fuel emissions, atmospheric growth rate and ocean sink (Friedlingstein et al., 2023). As the estimate of land carbon sink from DGVMs will introduce a budget imbalance in GCB2023, our estimates are well consistent with GCB2023's estimates based on ocean models and the residual land sink and close the global budget."

Specific comments:
1. *L25: Specify that these are in situ and flask $CO_2$ ObsPack data.*
**Response:**
    We revised the sentence to "The dataset was evaluated by comparing posterior $CO_2$ simulations with Total Carbon Column Observing Network (TCCON) retrievals as well as Observation Package (ObsPack) surface flask observations and aircraft observations." in Line 23-25.

2. *L102-103: I think Liu et al. (2021) optimized NBE, so may not be an applicable reference.*
**Response:**
    We revised the references to include only those inversions that optimize NEE.

3. *L115: "to December 21, 2022". Typically, inversions have a spin down period to increase data constraints at the end of the period, why was the inversion not extended into 2023?*
**Response:**
    GONGGA adopts the approach to optimize fluxes within each inversion window of 14 days. Once the fluxes were already optimized in previous windows, they will not change when the inversion window moves on. So we think that even a spin-down period

of the inversion till 2023 will not change our results for 2015-2022. Most studies using such an approach with limited length of inversion window usually do not include a spin down period (Jiang et al., 2022; Peters et al., 2007; Kong et al., 2022).

We agree with the reviewer that some inversions assimilating *in-situ* observations with the 4DVar algorithm and optimizing the full-time series of fluxes include a spin down period to account for the fact that the signal from a flux emitted in the Northern Hemisphere may take about 1 year to reach the Southern Hemisphere. However, we think that this issue may be of less concern when assimilating satellite $XCO_2$ observations compared to assimilating surface *in-situ* observations, given the fact that the wide coverage of satellite retrievals can capture the inter-hemisphere transport more easily and attribute the fluxes to its sources.

In addition, we show the time series of biases in the modelled $CO_2$ driven by posterior fluxes against the TCCON and ObsPack data in Fig. R3 and Fig. R4. At most stations, the biases in the year 2022 were similar to previous years and no significant trends were found, confirming that our estimates of fluxes in 2022 were not biased.

[Figure]

**Figure R3. Time series of monthly bias between TCCON retrievals and posterior simulations at each TCCON site (posterior simulation - retrieval).**

[Figure]

**Figure 4. Time series of monthly bias between ObsPack surface flask observations and posterior simulations for the six sub-regions (posterior simulation - observation).**

*4. Table 1 should be referenced in Sec. 2.4.1*
**Response:**
    We moved Table 1 to Sec. 2.4.1 and added the reference in the revised text as you recommended.

*5. L183: would be clearer to say "ocean-atmosphere" than "ocean"*
**Response:**
    Thank you for the suggestion to make the text clearer. We changed the "ocean flux" to "ocean-atmosphere flux" throughout the manuscript following your recommendation.

*6. L193-194: "NEE had substantial interannual variability (−4.08 ± 0.53 PgC yr⁻¹)". This phrasing makes it seem like −4.08 is the interannual variability. I would suggest re-phasing "NEE had substantial mean sink with considerable interannual variability, estimated as the standard deviation across years (−4.08 ± 0.53 PgC yr⁻¹)".*
**Response:**
    Thank you for the suggestion. We revised this sentence to "Over these 8 years, NEE exhibited a substantial mean sink with considerable interannual variability, estimated

as the standard deviation across years (–4.08 ± 0.53 Pg C yr$^{-1}$).” in Line 207-208.

*7.    L232: These are the incorrect citations for the v10 OCO-2 MIP. The documentation of the OCO-2 v10 MIP should be cited as:*
*Byrne, B., Baker, D. F., Basu, S., Bertolacci, M., Bowman, K. W., Carroll, D., Chatterjee, A., Chevallier, F., Ciais, P., Cressie, N., Crisp, D., Crowell, S., Deng, F., Deng, Z., Deutscher, N. M., Dubey, M. K., Feng, S., García, O. E., Griffith, D. W. T., Herkommer, B., Hu, L., Jacobson, A. R., Janardanan, R., Jeong, S., Johnson, M. S., Jones, D. B. A., Kivi, R., Liu, J., Liu, Z., Maksyutov, S., Miller, J. B., Miller, S. M., Morino, I., Notholt, J., Oda, T., O'Dell, C. W., Oh, Y.-S., Ohyama, H., Patra, P. K., Peiro, H., Petri, C., Philip, S., Pollard, D. F., Poulter, B., Remaud, M., Schuh, A., Sha, M. K., Shiomi, K., Strong, K., Sweeney, C., Té, Y., Tian, H., Velazco, V. A., Vrekoussis, M., Warneke, T., Worden, J. R., Wunch, D., Yao, Y., Yun, J., Zammit-Mangion, A., and Zeng, N.: National CO$_2$ budgets (2015–2020) inferred from atmospheric CO$_2$ observations in support of the global stocktake, Earth Syst. Sci. Data, 15, 963–1004, https://doi.org/10.5194/essd-15-963-2023, 2023.*
*While the dataset should be cited as:*
*Baker, D. F., Basu, S., Bertolacci, M., Chevallier, F., Cressie, N., Crowell, S., Deng, F., He, W., Jacobson, A. R., Janardanan, R., Jiang, F., Johnson, M. S., Jones, D. B. A., Liu, J., Liu, Z., Maksyutov, S., Miller, S. M., Philip, S., Schuh, A., Weir, B., Zammit-Mangion, A., and Zeng, N.: v10 Orbiting Carbon Observatory-2 model intercomparison project, NOAA Global Monitoring Laboratory [data set], https://gml.noaa.gov/ccgg/OCO2_v10mip/, last access:. XXX*
**Response:**
Thank you for pointing this out, we corrected the citations for the v10 OCO-2 MIP.

*8.    L245-248: It could be interesting to plot the GONGGA prior and OCO v10 MIP priors as a supplementary figure. Would be interesting if these differences are also present there.*
**Response:**
Thank you for your suggestion. We replaced Fig. S1 with the prior estimates from GONGGA and OCO-2 MIP, considering that the posterior GONGGA estimates were presented in Fig. 5. We also put it here as Fig. R5. It is shown that Boreal North America was a carbon source in GONGGA's prior, while it was a carbon sink in OCO-2 MIP prior. The inverted fluxes from GONGGA and OCO-2 MIP were both carbon sinks, but the size of the sink in GONGGA was smaller than OCO-2 MIP. It is similar for Northern Africa where GONGGA and OCO-2 MIP prior both estimated it as a carbon sink, while the inverted fluxes from GONGGA and OCO-2 MIP were both carbon sources. The larger carbon source from OCO-2 MIP aligned with the smaller prior carbon sink than GONGGA. The manuscript was revised as follows:
-    Line 260-270: "GONGGA showed good agreement with OCO-2 MIP inversions for most regions, and divergences occurred mainly in Boreal North America and

Northern Africa. The difference between GONGGA and OCO-2 MIP inversions may be related to the prior NBE adopted and retrieval pre-processing methods utilized. In Boreal North America, GONGGA's prior emerged as a carbon source, whereas OCO-2 MIP's prior was a carbon sink (Fig. S1). After assimilating OCO-2 retrievals, GONGGA and OCO-2 MIP consistently showed Boreal North America was a carbon sink, but the sink in GONGGA was smaller than OCO-2 MIP.   The same situation happened in Northern Africa. Both GONGGA's prior and OCO-2 MIP's prior estimated Northern Africa as a terrestrial carbon sink, but the sink from GONGGA was stronger than that from OCO-2 MIP (Fig. S1). Constrained by OCO-2 retrievals, both GONGGA and OCO-2 MIP estimated it as a carbon source, and the source from GONGGA was weaker than that from OCO-2 MIP, aligning with the sizes of their prior sinks. In addition, the impact of prior fluxes may be amplified by the insufficient coverage of OCO-2 retrievals."

[Figure]

**Figure R5. Annual mean (2015–2022) NBE at 11 TransCom land regions from GONGGA prior and OCO-2 MIP prior estimates. The error bar of NBE represents the multi-year standard deviation.**

*9.  L258-259: I don't understand the logic in this sentence: "In the Amazon, the mean gross emissions from forest fires from 2003 to 2015 was $454 \pm 496$ Tg CO₂ yr⁻¹, which may counteract the decline of Amazon deforestation carbon emissions (Aragão et al., 2018)."*

**Response:**

To make things clearer, we moved this paragraph to the discussion section and rewrote this sentence as: "In the Amazon, despite the decline of deforestation rate during 2003-2015, carbon emissions from drought-induced fires had increased very quickly (Aragão et al., 2018)." in Line 398-400.

10. *L260: In addition to van der Velde et al. (2021), there were two studies that examined the CO₂ emissions from the 2019-20 Australian fires using OCO-2 data:*
*1. Byrne, B., Liu, J., Lee, M., Yin, Y., Bowman, K. W., Miyazaki, K., et al. (2021). The carbon cycle of southeast Australia during 2019–2020: Drought, fires, and subsequent recovery. AGU Advances, 2, e2021AV000469. https://doi.org/10.1029/2021AV000469*
*2. Wang, J., Liu, Z., Zeng, N., Jiang, F., Wang, H., & Ju, W. (2020). Spaceborne detection of XCO2 enhancement induced by Australian mega-bush-fires. Environmental Research Letters, 15(12), 124069. https://doi.org/10.1088/1748-9326/abc846*
**Response:**
 Thank you for mentioning these references. We added them in the text as recommended.

11. *L272: Please be more specific. I suggest re-writting "the magnitude of global NBE IAV" as "the standard deviation of global NBE IAV".*
**Response:**
 Indeed, we calculated the NBE IAV as the standard deviation of NBE across years. To be clearer, we revised this sentence to "We computed the standard deviation of global NBE to represent its magnitude of IAV, which amounted to 0.63 Pg C yr$^{-1}$ during the 2015–2022 period." in Line 282-284.

12. *L276-277: "Considering the short time series of the carbon cycle, the latitudinal contributions in this study are qualitative, rather than quantitative." I think this would be better written as "Considering the short time series of the carbon cycle, the latitudinal contributions in this study are suggestive but not statistically robust."*
**Response:**
 We revised this sentence to "Given the short time series of the inversion, the latitudinal contributions in this study are suggestive but not statistically conclusive." in Line 288-289.

13. *L298: "more flatten" should be "smaller amplitude"*
**Response:**
 Thank you for the suggestion. We revised the sentence to "In the tropics, however, the seasonal cycles have smaller amplitudes and the shapes are distinct in different years." in Line 308.

14. *Figure 9-12 captions. Specify "posterior simulations"*
**Response:**
 We added "posterior simulations" in Figure 9-12 captions and explained in the text that it means "the simulation is driven by posterior fluxes" in Line 333.

*15. L363: Just for your information, there is a known difference in the mean atmospheric CO₂ abundance between TCCON and posterior CO₂ fields from in situ inversions, which is not well understood. I'm not sure if this has been documented in a paper, but it is known to some researchers. This could cause the differences seen here.*
**Response:**

Thank you for the information. We found in Polavarapu et al., (2018) and Peiro et al., (2022) that the posterior $CO_2$ simulations from *in-situ* inversions exhibited high positive biases relative to TCCON retrievals in northern mid- to high- latitudes. We will keep on following the latest studies.

*16. Q: L366: BIAS shouldn't be all capitalized.*
**Response:**

Thank you for the suggestion. We changed all "BIAS" to "bias".

**References**

[revised manuscript text omitted]

---

## Author Comment (AC2)

**A global surface CO₂ flux dataset (2015–2022) inferred from OCO-2 retrievals using the GONGGA inversion system**

**Manuscript No. essd-2023-449**

Zhe Jin, Xiangjun Tian, Yilong Wang, Hongqin Zhang, Min Zhao, Tao Wang, Jinzhi Ding, Shilong Piao

**Reply to reviewer 2**

We would like to thank reviewer 2 for the constructive suggestions and insightful comments, which have helped us improve the manuscript. Please see below the point-to-point responses. All the line numbers indicated in this letter of response correspond to the revised version of text.

**Reviewer 2:**
*The authors introduce a new inversion system (GONGGA) that assimilates total column CO₂ from NASA's OCO2 satellite to optimize terrestrial and oceanic carbon fluxes (NEE). The results are compared against a recent model inter-comparison project that assimilated an older version of this dataset. Results are also evaluated against a network of upward looking forward scattering radiometers as well as in-situ surface and aircraft observations.*
*The manuscript adds a novel inversion system to a growing list of similar simulations (global models that estimate CO₂ fluxes by assimilating total column CO₂ retrievals). The manuscript is generally well written. Therefore, I think this is suitable for publication in ESSD.*

**Response:** We thank the reviewer for the positive evaluation of our study and the valuable suggestions to improve it. We have carefully revised our manuscript following the comments and suggestions.

I only have a few concerns at this point:

*1. Q: It isn't clear how XCO₂ uncertainties are treated in the inversion system. It is generally assumed that the reported XCO₂ uncertainty in the lite files is likely too low. Moreover, unlike in-situ observations, XCO₂ data exhibit high correlation (given that individual soundings are only 300 m apart). Therefore, the information content as well as errors are highly correlated for adjacent soundings. Generally, studies have relied on averaging. See Piero et al. 2022, Byrne et al. 2023, and Baker et al., 2022. I would recommend expanding the methods section to describe exactly how retrieval uncertainties are treated (given the context of the afore-mentioned studies) and perform some sensitivity analyses (e.g., tests where uncertainties are inflated) to estimate the impact of data error on retrieved fluxes.*

**Response:**

Thank you for mentioning the detailed procedure applied by the OCO-2 MIP project. In this inversion experiment, we used a different approach than the "super-obs" approach of the OCO-2 MIP project as explained below.

We applied an observation thinning algorithm to reduce the number of observations. Observation thinning is usually used in data assimilation for numerical weather prediction (NWP), and is efficient in reducing the error-correlation (Liu and Rabier, 2002; Campbell et al., 2017; Reale et al., 2018). During the whole period from September 6, 2014, to December 31, 2023, only one fifth of the total $XCO_2$ retrievals were assimilated. We added the reference for the rationality of observation thinning and explained the procedure in the revised Sec. 2.3:

- Line 147-153: "We applied a data thinning algorithm (Liu and Rabier, 2002; Campbell et al., 2017; Reale et al., 2018) to reduce the potential impacts of correlated errors in adjacent soundings. We set the threshold of the number of daily observations to 20,000. If the number of good retrievals exceeded the threshold within a single day, excess data were removed. For example, if there were 60,000 good retrievals in one day, one of every three sequential retrievals was selected according to sounding ID. Before data thinning, there were 203,368,424 $XCO_2$ retrievals with good quality from September 6, 2014, to December 31, 2022. After data thinning, 40,337,763 $XCO_2$ retrievals were actually assimilated in the inversion, about a fifth of total good retrievals."

We are now working on adapting our system to use the "super-obs" approach following the OCO-2 MIP protocol, and will probably publish the results in the next version of GONGGA. We added a paragraph in the discussion section to outline the future development of GONGGA to align with the latest knowledge from OCO-2 science team:

- Line 422-427: "In addition, some parts of GONGGA's inversion algorithm, such as the data selection, were partly different from those proposed by the OCO-2 Science Team (Peiro et al., 2022; Byrne et al., 2023; Baker et al., 2022), but GONGGA's inversion results were broadly consistent with the ensemble of OCO-2 MIP inversions and GCB2023, and gave reasonable estimates of global and regional carbon budgets within the uncertainties. In the future, GONGGA will regularly publish new versions of inverted fluxes using the latest OCO-2 data on an annual basis. These updates will align with the latest suggestions from the OCO-2 Science Team, enabling the ongoing monitoring of $CO_2$ fluxes."

For the concern about the retrieval uncertainty, we used the xco2_unvertainty reported in the OCO-2 Lite file. To investigate the impacts of data error on inverted fluxes, we conducted three sensitivity tests. In the first and second experiments (E1 and E2), the reported $XCO_2$ uncertainties were enlarged by two and four folds, respectively. In the third experiment (E3), the $XCO_2$ uncertainties were added by 5 ppm. The three sensitivity tests adopted the same set-ups as the inversion in this study only except for the $XCO_2$ uncertainties. At the global scale, the inverted annual NBE and $F_{OCEAN}$ from the original inversion, E1, and E2 are very close, but E3 has a different portioning between land and ocean fluxes than the other inversions, which amounts to about 0.2

Pg C yr$^{-1}$ (Fig. R6). When it comes to the regional scale, the differences are larger in some regions and years but are still broadly consistent with the reference inversion (Fig. R7). In the revised manuscript, we added the results of these sensitivity tests and discussed the impacts of data errors:

- Line 406-416: "The processing of $XCO_2$ uncertainties also had an impact on the inversion results. We performed three sensitivity inversions with different $XCO_2$ uncertainties. The $XCO_2$ uncertainties were inflated two and four times in the first (E1) and second (E2) test, respectively. In the third test (E3), the $XCO_2$ uncertainties were increased by 5 ppm. The three sensitivity tests adopted the same configuration as the reference inversion in this study only except for the $XCO_2$ uncertainties. The distributions of different $XCO_2$ uncertainties were shown in Fig. S8. At the global scale, the inverted annual NBE and $F_{OCEAN}$ from the original inversion, E1, and E2 were very close, but E3 had a different partitioning between land and ocean fluxes than the other inversions, which amounted to about 0.2 Pg C yr$^{-1}$ (Fig. S9). When it comes to regional scale, the differences were larger in some regions and years but were still broadly consistent with the reference inversion (Fig. S10). These sensitivity tests highlighted the fact that the inversion results were indeed impacted by the assumption regarding $XCO_2$ uncertainty and careful assessment of uncertainties in satellite $XCO_2$ retrievals was necessary for accurate estimates of global and regional carbon fluxes."

[Figure]

**Figure R6. The global annual NBE and $F_{OCEAN}$ from GONGGA posterior estimates with default OCO-2 v11r $XCO_2$ uncertainties (orange), doubled OCO-2 v11r original $XCO_2$ uncertainties (green), quadrupled OCO-2 v11r original $XCO_2$ uncertainties (purple), and OCO-2 v11r original $XCO_2$ uncertainties added by 5 ppm (yellow).**

[Figure]

**Figure R7. NBE in 11 TransCom land regions from GONGGA posterior estimates with default OCO-2 v11r XCO₂ uncertainties (orange), doubled OCO-2 v11r original XCO₂ uncertainties (green), quadrupled OCO-2 v11r XCO₂ uncertainties (purple), and OCO-2 v11r XCO₂ uncertainties added by 5 ppm (yellow).**

2.  *Q: The authors note that the main differences from the OCO v10 MIP arise in the high northern latitudes. At one point this was due an issue with the OCO-2 retrievals in the v11 dataset. I wonder if the retrievals used in the inversion system are impacted by this. I would check with the dataset providers to see if the authors are using a version that is known to have issues. Also see the data quality statement: https://docserver.gesdisc.eosdis.nasa.gov/public/project/OCO/OCO2_L2_ Data_Release_Statement_v11.1_Lite_Files.pdf*

**Response:**

We noticed that in v11r Lite dataset, the XCO₂ data may have a bias of about -0.4 ppm due to the impact of a different DEM data used in the retrieval algorithm (Jacobs, et al., 2024). It was found that this negative bias would introduce a larger sink in the northern high latitudes. To check the impact of such biases in XCO₂, we run the inversion with the latest v11.1r product for the period from 2015 to 2018. The results show that the inverted sink north of 60°N using v11.1r are smaller than that using v11r by 90 to 140 Tg C yr⁻¹, and accompanied by a compensating increase in ocean carbon uptake in the northern mid- and low- latitudinal band (Fig. R8). That means the sink in Boreal North America from GONGGA will be even smaller than that from the OCO-2 MIP if v11.1r data are used. Thus, the difference between GONGGA and OCO-2 MIP in the estimates of sink in Boreal North America seems not due to the negative biases in v11r Lite XCO₂ retrievals but is likely due to other differences, such as the inversion

algorithm, the prior fluxes, and the associated prior uncertainty. In particular, we found that Boreal North America was a carbon source in the prior flux of GONGGA, while it was a carbon sink in the OCO-2 MIP prior. After assimilating the observations, Boreal North America emerged as a carbon sink in both GONGGA and OCO-2 MIP, but the sink size was smaller in GONGGA than OCO-2 MIP. In the revised manuscript, we added a paragraph to discuss the potential impacts of using different versions of OCO-2 retrieval products on inverted fluxes, and emphasized that we would regularly update the GONGGA results with the latest OCO-2 dataset:

- Line 417-427: "In the current version of GONGGA, we assimilated the OCO-2 v11r Lite $XCO_2$ dataset. A recent paper found that the v11r Lite product has a bias of -0.4 to -0.8 ppm across regions north of 60°N due to the variations of digital elevation model (DEM) used in the retrieval algorithm (Jacobs et al., 2024), and this bias introduces a ~ 100 Tg C shift in the partitioning of carbon fluxes for the latitudinal bands. A preliminary test of GONGGA using the latest v11.1r Lite product showed the inverted terrestrial carbon sink tends to be 90 to 140 Tg C $yr^{-1}$ lower north of 60° N than using the v11r Lite product, consistent with the previous findings. In addition, some parts of GONGGA's inversion algorithm, such as the data selection, were partly different from those proposed by the OCO-2 Science Team (Peiro et al., 2022; Byrne et al., 2023; Baker et al., 2022), but GONGGA's inversion results were broadly consistent with the ensemble of OCO-2 MIP inversions and GCB2023, and gave reasonable estimates of global and regional carbon budgets within the uncertainties. In the future, GONGGA will regularly publish new versions of inverted fluxes using the latest OCO-2 data on an annual basis. These updates will align with the latest suggestions from the OCO-2 Science Team, enabling the ongoing monitoring of $CO_2$ fluxes."

We also expanded the discussion about the regional differences in inverted fluxes between GONGGA and OCO-2 MIP in Sec. 4.2:

- Line 260-269: "GONGGA showed good agreement with OCO-2 MIP inversions for most regions, and divergences occurred mainly in Boreal North America and Northern Africa. The difference between GONGGA and OCO-2 MIP inversions may be related to the prior NBE adopted and retrieval pre-processing methods utilized. In Boreal North America, GONGGA's prior emerged as a carbon source, whereas OCO-2 MIP's prior was a carbon sink (Fig. S1). After assimilating OCO-2 retrievals, GONGGA and OCO-2 MIP consistently showed Boreal North America was a carbon sink, but the sink in GONGGA was smaller than OCO-2 MIP. The same situation happened in Northern Africa. Both GONGGA's prior and OCO-2 MIP's prior estimated Northern Africa as a terrestrial carbon sink, but the sink from GONGGA was stronger than that from OCO-2 MIP (Fig. S1). Constrained by OCO-2 retrievals, both GONGGA and OCO-2 MIP estimated it as a carbon source, and the source from GONGGA was weaker than that from OCO-2 MIP, aligning with the sizes of their prior sinks."

[Figure]

**Figure R8. The inverted (a) NBE and (b) $F_{OCEAN}$ in regions north of 60°N and 0-60°N from GONGGA inversions using OCO-2 v11r retrievals (orange bars) and v11.1r (green bars) retrievals during 2015-2018.**

Minor comments:

*1. Lines 99-100: Biomass burning carbon emissions are also terrestrial ecosystem fluxes, so I would just define NEE instead (i.e., balance of photosynthesis and respiration).*

**Response:**

Thank you for the suggestion. We used NEE in the revised manuscript.

*2. Line 153- Cite ObsPack and also specify which version was used.*

**Response:**

The ObsPack data we used were $CO_2$ GLOBALVIEW plus v8.0 and NRT v8.1. Version information and citations were added in Line 164-165.

*3. Line 176-178: CARVE aircraft observations may not be appropriate for evaluation, given that CARVE flight tracks did not intend to sample regions that were representative of large areas. I would recommend removing CARVE, or discussing this when you discuss results for Fig. 12.*

**Response:**

We removed CARVE aircraft observations in our evaluations and added data from NASA's ATom Mission.

*4. Line 190-92: Earlier it was stated that fossil fuel and biomass burning $CO_2$ emissions were not optimized. So it would be incorrect to say that $E_{FOS}$ and $E_{FIRE}$ were quantified. Instead they were specified.*

**Response:**

Thank you for the suggestion. We included the fossil fuel and biomass burning $CO_2$ emissions, although not optimized, in the dataset for readers who may be interested in all the carbon budgets. To be clearer, we revised the sentence: "Here, we present the five major components of the global carbon budget, including the fossil fuel $CO_2$ emissions ($E_{FOS}$), biomass burning emissions ($E_{FIRE}$), atmospheric $CO_2$ concentration growth rate ($G_{ATM}$), ocean-atmosphere carbon fluxes ($F_{OCEAN}$), and NEE." in Line 203-205.

*5. Line 192: $S_{LAND}$ generally refers to NEE + BMB fluxes. I would suggest changing this to NEE throughout. You could then call the sum of NEE and BMB $S_{LAND}$ or NBE.*

**Response:**

Thank you for the suggestion. We agree that the Global Carbon Budget report also used $S_{LAND}$ and $S_{OCEAN}$, but they mainly represent the carbon stock changes, which are different from the inverted fluxes. To be clearer, we changed the notations in the revised manuscript. We used NEE for the inverted land fluxes excluding biomass burning and fossil fuel emissions, NBE for the sum of NEE and biomass burning emissions, and $F_{OCEAN}$ for the inverted atmosphere-ocean fluxes.

*6. Line 231: The OCO-MIP V10 citations should be Byrne et al., (2023). I think here it should also be noted that the v10 MIP assimilated v10 OCO-2 retrievals while in this study OCO-2 v11r retrievals are assimilated.*

**Response:**

Thank you for the correction. We corrected the citations for OCO-MIP v10 and revised the sentence to: "Here, we present the GONGGA-estimated annual mean (2015–2022) NBE for 11 TransCom land regions and their comparison with OCO-2 model intercomparison project (MIP) v10 inversions (Fig. 5). OCO-2 MIP v10 (Byrne et al., 2023; Baker et al., 2023) includes an ensemble of 14 atmospheric inversions over the period of 2015–2020 assimilating OCO-2 v10r retrievals, and each of them is characterized by distinct transport models, data assimilation algorithms, and prior fluxes (Table S1)." in Line 250-254.

*7. Lines 258-263: This is citing previous work and should go in a discussion section rather than the results, since it reads like it is a result of this study, which it is not. Finally, given that biomass burning fluxes were not optimized, I think these should be discussed in terms of how their magnitude is relative to the NEE fluxes (that were optimized).*

**Response:**

Thank you for your suggestion. We moved the paragraph to the discussion section and revised the text:

- Line 394-405: "Regarding the regional carbon budget, we found that fire emission, although it was not optimized in the inversion, largely impacted the net $CO_2$ fluxes from terrestrial ecosystem, i.e. NBE, in equatorial regions and Australia. With the frequent occurrence of wildfires in these regions, carbon emissions from wildfires may exceed regional NEE and make these regions net carbon sources (Fig. S7). For the past few decades, ~50% of fire-related carbon emissions and ~70% of global burned areas occurred across African subtropical savannah systems (Andela and Van Der Werf, 2014; Giglio et al., 2013). In the Amazon, despite the decline in deforestation rate during 2003-2015, carbon emissions from drought-induced fires had increased very quickly (Aragão et al., 2018). Southeast Australia also experienced intensive and geographically extensive wildfires during the 2019–2020 summer season, and the fires released substantial amounts of $CO_2$ into the atmosphere (Van Der Velde et al., 2021; Byrne et al., 2021; Wang et al., 2020). As a result, the 8-yr mean biomass burning emissions in Tropical South America, Northern Africa, and Tropical Asia amounted to 0.17, 0.33, and 0.13 Pg C $yr^{-1}$, and were 6.2, 1.2, and 1.4 times higher than regional NEE, respectively, resulting in net carbon sources in these regions. The increasing fire emissions thus present a great challenge to climate mitigation efforts."

*8.    Fig. 6: The caption for this figure should say NBE, not IAV of NBE, since each point on the figure refers to a value not an IAV. Also it seems from this figure that most of the IAV comes from North Extra tropics, but in lines 273-76 the authors say that tropics contribute 100% to global IAV. I think this should be clarified.*

**Response:**

We changed the caption for Fig. 6 as "Annual NBE over the globe, northern extra-tropics (30–90°N), tropics (30°S–30°N), and southern extra-tropics (90–30°S) during 2015–2022." Figure 6 shows that although the total global land carbon sink was mainly located in the northern extra-tropics (green line), the shape of global year-to-year fluctuations (black line) tightly follows that of the tropics (red line). For example, the enhanced global NBE in 2017 and 2022 compared to other years was mainly caused by an increased NBE in the tropics, while the NBE in the northern extra-tropics in these two years was close to other years. In addition, using Eq. (1) from Ahlström et al. (2015), which accounts for the synchronicity of the sign and magnitude in the NBE change between regions and the globe, we found the tropics contribute about 100% of the global NBE IAV. To be clearer, we changed this figure to show the NBE anomalies in different regions rather than the absolute values.

*9.    Q: In all figures with labels "PgC" should be changed to "Pg C".*

**Response:**

We changed all the "PgC" and "PgC $yr^{-1}$" to "Pg C" and "Pg C $yr^{-1}$" in the figures,

as suggested.

*10. Line 317: Observed XCO₂ should be changed to retrieved XCO₂ given that XCO₂ cannot be observed directly.*

**Response:**

We changed the "observed XCO₂" to "TCCON XCO₂ retrievals" in this sentence and other places where necessary.

*11. Figures 11 and 12. RMSE folds in both random and systematic error (bias). See Rastogi et al., (2021) for discussion of bias and random error evaluation of OCO-2 relative to in-situ aircraft observations. For vertical profile data, I think it would be useful to look at errors in the column. For instance, the model may have errors that cancel in the column (e.g., high bias near the surface and a low-bias aloft).*

*Instead, I would recommend the authors to report random error and bias separately. Also, why is bias capitalized?*

**Response:**

Thank you for the suggestion. We separately reported biases and random errors in the revised manuscript.

Regarding the vertical profile, we added the evaluations in Fig. S6 in the revised manuscript, which is also shown here as Fig. R9. The random errors, which were quantified by the standard deviation of the mismatch between observations and modelled $CO_2$, were typically within 1 ppm. Large biases, up to 2 ppm, were seen in the high latitudes and above 9 km. This is consistent with the comparisons against the TCCON observations (Fig. 8 in the manuscript). The reason for such large biases can be attributed to the underestimation of land carbon sink in the high latitudes as compared to OCO-2 MIP, or the biases in the atmospheric transport. We added the following texts to the revised manuscript:

- Line 385-388: "We also compared the vertical distribution of modelled $CO_2$ against the observations. Figure S6 shows that the random errors were typically within 1 ppm, showing a good agreement between the simulations and observations. However, large biases, up to 2 ppm, were seen in the high latitudes and above 9 km, consistent with the comparisons against the TCCON retrievals (Fig. 8)."

[Figure]

**Figure R9. (a) random error and (b) bias between posterior CO₂ simulations and aircraft observations as a function of latitude and altitude (posterior simulations minus observations; unit: ppm). The altitudes are binned every kilometer from 1 km to 12 km, and for altitudes above 12 km.**

*12. In this section there are other datasets such as the atmospheric tomography mission (ATom) aircraft campaigns would have been valuable for evaluation. See Gaubert et al., (2023) for details. I would also advise the authors to look at the OCO MIP website for evaluation against observations (surface and partial columns).*
**Response:**
Thank you for the suggestion. We added the atmospheric tomography mission (ATom) aircraft data to our evaluation. We changed Fig. 1(b) to include the distribution of ATom observations, and Fig. 12 in the revised manuscript gave the evaluation including Atom observations.

**References**

[revised manuscript text omitted]